# Diffusion Active Learning: Towards Data-Driven Experimental Design in Computed Tomography

## Abstract

We introduce *Diffusion Active Learning*, a novel approach that integrates a generative diffusion model with sequential experimental design to adaptively acquire data for solving inverse problems in imaging. We first pre-train an unconditional diffusion model on domain-specific data. The diffusion model is aimed to capture the structure of the underlying data distribution, which is then leveraged in the active learning process. During the active learning loop, we use the forward model of the inverse problem together with the diffusion model to generate conditional data samples from the posterior distribution, all consistent with the current measurements. Based on the generated samples we quantify the uncertainty in the current estimate in order to select the most informative next measurement. We showcase the proposed approach for its application in X-ray computed tomography imaging. Our results demonstrate significant reductions in data acquisition requirements (*i.e.*, lower X-ray dose) and improved image reconstruction quality across several real-world tomography datasets.

## 1 Introduction

Computed Tomography (CT) is an imaging technique for reconstructing objects from X-ray projection data. The concept of tomography originated with the work of Radon in 1917, however, it was not until 1971 that Godfrey Hounsfield and Allan Cormack developed the first practical CT scanner. Recent advances at large synchrotron facilities have pushed CT resolution into the nanometer range, enabling novel scientific applications, such as the inspection of composite materials, quality control of computer chips, and revealing cellular structure in biological tissues. Achieving a resolution as low as several nanometers requires image acquisition times of up to several days (Aidukas et al., 2024). In such settings, the X-ray dose deposited onto the sample becomes a key limiting factor, causing radiation damage which ultimately limits the achievable resolution (Howells et al., 2009). Learning based reconstruction methods and data-driven 'smart' acquisition techniques are a promising avenue to improve data efficiency, allowing for high-resolution reconstructions with lower acquisition times and reduced X-ray dose.

Mathematically, tomographic projections are described by the Radon transform (Deans, 1983) of the object. In the corresponding inverse problem, multiple 2D projections from different angles are combined into a single 3D reconstruction of the object (Fig. 1). However, traditional reconstruction algorithms such as filtered back-projection or iterative reconstruction schemes only make use of information contained in the measurements (Kak & Slaney, 2001), and neglect additional structure in the data distribution. For example, tomographic scans of computer chips or composite materials display highly regular structures (see Fig. 3). Such regularity can be learned from prior data sets and facilitate improved learning-based reconstruction algorithms.

Moreover, standard CT acquisition scans are acquired by either rotating the object or the CT scanner over an equidistant angle grid. Uniform scanning neglects any potential structure in the data distribution, that could allow to achieve better reconstructions by scanning the most informative angles. Active learning is a subfield of machine learning that studies algorithms for adaptive data acquisition, with the goal to obtain the most informative data points (Settles, 2009). However, the adoption of active learning in applications remains challenging, with literature highlighting instances

where active learning methods have shown limited effectiveness compared to conventional uniform or i.i.d. training schemes (e.g., Lowell et al., 2019). This is due to the many underlying challenges in sequential decision-making algorithms, such as the need for uncertainty quantification, as well as added computational and implementation complexity. The effectiveness of active learning further relies on exploiting inherent structure in the data distribution and on statistical modeling assumptions (Balcan et al., 2010), which are often difficult to specify for the domain of interest.

In this work, we set to address the outlined challenges in computed tomography by combining a learned generative prior with data-driven experimental design. We introduce *Diffusion Active Learning* (DAL), which combines generative diffusion models with active learning to solve inverse problems in imaging. To leverage the structure of the data distribution, we pre-train the diffusion model on data collected from slices of tomographic reconstructions (*e.g.*, image slices of integrated circuits or composite materials). In the active learning loop, we generate samples from the posterior distribution of the diffusion model, conditioned on the measurements collected so far. Based on the generated samples, we quantify the uncertainty in the solution of the inverse problem, and use it to select where to sample next. By repeating this process, and as more measurements are incorporated, we effectively constrain the Diffusion posterior distribution until it collapses to a deterministic, data-consistent final estimate.

Using a diffusion model as a prior for the active learning enables several key advantages: $(i)$ Unlike traditional regularizers (*e.g.*, Total Variation), the learned prior is data-dependent and captures problem specific structure. $(ii)$ The structure can be extracted from (large-scale) prior data, avoiding to manually specify intricate regularities in the statistical modeling approach. $(iii)$ The diffusion model successfully captures multi-modal, highly structured distributions (*e.g.*, natural images). This is unlike, for example, the Gaussian Laplace approximation used in prior works (Antoran et al., 2023; Barbano et al., 2022a), which are inherently unimodal.

We demonstrate the effectiveness of our approach on three real-world tomography datasets and conduct an extensive evaluation of different acquisition strategies. Our results shows that DAL provides significant improvements in reconstruction quality with fewer measurements compared to conventional uniformly sampled projection measurements. For the scientific datasets benchmarked in this paper, we achieve the same average Peak Signal-to-Noise Ratio (PSNR) in the reconstruction with up to four times fewer measurements, which translates to up to $4\times$ reduction in X-ray dose (see Table 1). Furthermore, we achieve this with more than $2\times$ computational speed-ups compared to the second-fastest baseline (Barbano et al., 2022a) (see Figure 5)

**Remark:** In a sparse reconstruction settings, diffusion-based approaches can introduce artifacts or hallucinations as not enough data is measured; preventing a direct use in high-stakes setting such as medical diagnosis (medical X-ray scans are also very fast, and therefore not considered a direct application of the proposed approach). The sparse reconstruction setting, however, is inherently ill-posed, and as such, a learned prior provides a good trade-off between provable accuracy and data-efficiency.

## 1.1 RELATED WORK

There is a long history of work that uses ideas from computer vision to improve image quality in tomography and medical imaging, see *e.g.*, (Li et al., 2022; Parvaiz et al., 2023) for comprehensive surveys. Building on ideas from image denoising and segmentation (Ronneberger et al., 2015), Ulyanov et al. (2018) propose the *Deep Image Prior* (DIP), which utilizes the structure of a randomly initialized convolutional network (U-net) as a prior for image reconstruction tasks. This approach demonstrated that even without pre-training, deep neural network architectures can provide an implicit bias that improves image reconstruction quality. The DIP methodology has influenced various subsequent works that incorporate deep learning architectures in inverse problem settings, *e.g.*, using pre-training and an initial reconstruction (Baguer et al., 2020; Barbano et al., 2022b).

The success of diffusion-based models in image reconstruction has also been extended to solve inverse problems. When working with ill-posed inverse problems and using sparse measurements, the goal is to use the diffusion model as a prior to fill in the missing information in the reconstruction. However, sampling from the posterior distribution conditioned on the measurements is a challenging task. As the number of measurements may be different every time, techniques for conditional sampling used in common text-to-image model have not gained traction for inverse problem solv-

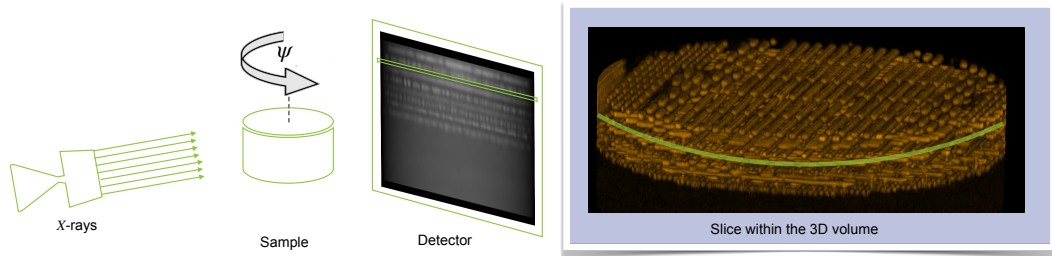

Figure 1: Left: An illustration of X-ray tomography with rotated sample and projection measurements on the detector. Right: 3D reconstruction can be done slice by slice of each vertical sample layer, simplifying tomographic reconstruction to a series of 2D reconstructions from 1D projections.

ing. Nonetheless, Song et al. (2021) apply score-based generative models specifically to sparse reconstruction in medical CT imaging. They sample from the posterior distribution by steering the diffusion process using the measurements. This decouples the training and inference processes and is adopted by most methods hereafter. Their technique is however limited to linear models and tends to fail as measurements become noisy. Chung et al. (2022) introduced the concept of *Diffusion Posterior Sampling*. By modifying the reverse diffusion process, they sample from the posterior distribution, significantly enhancing the reconstruction quality from incomplete or noisy data. Song et al. (2023) extend the use of diffusion models by introducing latent space diffusion models and the concept of *Hard Data Consistency*. This approach solves an optimization problem to align generated samples with observed data, thereby ensuring that reconstructions adhere closely to the measurements. Their method shows notable improvements in handling ill-posed inverse problems, particularly in medical imaging contexts. Lastly, Barbano et al. (2023) proposed a steerable conditional diffusion model designed to adapt to out-of-distribution scenarios in imaging inverse problems. Their method ensures that the generative process remains robust even when the data deviates from the training distribution.

Active learning is a huge field (*e.g.*, Settles, 2009) with roots in Bayesian experimental design (Chaloner & Verdinelli, 1995). The active learning literature has mostly focused on the classification setting, where the goal is to reduce the labelling effort by actively querying the label for the most uncertain data points. These ideas have been applied to deep learning Ren et al. (2021) and medical imaging Budd et al. (2021) as well. Gal et al. (2017) discuss various acquisition strategies for active learning with image data, also applicable in the regression setting relevant to this work. Only few prior works have explored experimental design for computed tomography. Closely related is the work by Barbano et al. (2022a), who introduce Bayesian experimental design for computed tomography. Their main innovation is the use of a Linearized Laplace Deep Image Prior for uncertainty quantification Antoran et al. (2022) to guide the acquisition of measurement data. Although this demonstrates the applicability of sequential experimental design in the CT setting, the evaluation is still limited to a simplistic toy example. Subsequently, Antoran et al. (2023) further scale the Laplace approach using a sampling based technique and demonstrate it on a CT reconstruction task, although not in combination with active learning.

Going well-beyond one-step (greedy) active learning schemes, reinforcement learning (RL) provides a framework to solve a multi-stage planning problem over the combinatorial space of possible experimental designs. This is prominently explored in the related field of magnetic resonance imaging (MRI), see, for example, the works by Zhang et al. (2019); Pineda et al. (2020); Bakker et al. (2020); Jin et al. (2019). Similarly, RL based methods for computed tomography are proposed by Wang et al. (2023); Shen et al. (2022). These approaches primarily address the problem of choosing an optimal sequence of measurements for a given reconstruction method; this comes at the price of increased algorithmic and training complexity compared to greedy active learning schemes. We also remark that unlike MRI, the tomographic forward model is linear, and greedy approaches provably converge (Riquelme et al., 2017). In principle, RL based methods can be combined with a learned prior; this is however beyond the scope of this work.

---

**Algorithm 1:** Diffusion Active Learning

---

**Input:** $k$ : number of conditional samples, $\mathcal{D}_1$ : initial set of measurements, $\mathcal{M}$ : pre-trained diffusion model

1 **for** $t = 1, \ldots, n$ **do**
2      samples: $\boldsymbol{x}_t^1, \ldots, \boldsymbol{x}_t^k = \mathcal{M}.\texttt{conditional\_sampling}(\texttt{data} = \mathcal{D}_t, \texttt{num\_samples} = \texttt{k})$
3      mean prediction: $\bar{\boldsymbol{x}}_t = \frac{1}{k} \sum_{i=1}^k \boldsymbol{x}_t^i$
4      maximum variance acquisition: $\psi_t = \arg\max_{\psi \in \Phi} \frac{1}{k} \sum_{i=1}^k \|\mathcal{A}_\psi(\boldsymbol{x}_t^i) - \mathcal{A}_\psi(\bar{\boldsymbol{x}}_t)\|^2$
5      new measurement: $\boldsymbol{y}_{\psi_t} = \mathcal{A}_{\psi_t}(\boldsymbol{x}^*) + \epsilon_t$
6      data update: $\mathcal{D}_{t+1} = \mathcal{D}_t \cup \{(\psi_t, \boldsymbol{y}_{\psi_t})\},$

---

## 2 SETTING

In this work, we consider the reconstruction of 2D objects (images) from their 1D projections (measurements). The reconstruction of 3D volumes from 2D projections follows from the same principle. Alternatively, 3D volumes can also be reconstructed by stacking 2D slices. Figure 1 illustrates a typical acquisition setup for 3D volume reconstruction.

Formally, let $\boldsymbol{x}^* \in \mathbb{R}^{d \times d}$ be a 2-dimensional grayscale image, corresponding to a slice of the object that we aim to reconstruct. In our simplified setup, we assume that the detector has $l \in \mathbb{N}$ pixels, corresponding to the resolution of the observed projection. For a given angle $\psi$, the observed measurement $\boldsymbol{y}_\psi \in \mathbb{R}^l$ is given by a forward operator $\mathcal{A}_\psi : \mathbb{R}^{d \times d} \to \mathbb{R}^l$ and a noise vector $\epsilon \in \mathbb{R}^l$,

$$\boldsymbol{y}_\psi = \mathcal{A}_\psi(\boldsymbol{x}^*) + \epsilon. \tag{1}$$

The noise is often assumed to be Gaussian or Poisson distributed. For the special case of parallel beam tomography, the forward model $\mathcal{A}_\psi(\boldsymbol{x})$ is a linear operator that is mathematically determined by the Radon transform (Kak & Slaney, 2001).

For a given set of $n$ projections $\boldsymbol{\mathcal{Y}}_{\boldsymbol{\psi}} = \{\boldsymbol{y}_{\psi_1}, \ldots, \boldsymbol{y}_{\psi_n}\}$ (or $\boldsymbol{\mathcal{Y}}$ for simplicity) and measured at angles $\boldsymbol{\psi} = \{\psi_1, \ldots, \psi_n\}$, the goal is to reconstruct $\boldsymbol{x}$ from $\boldsymbol{\mathcal{Y}}$. Assuming a Gaussian distribution of the noise $\epsilon$, the reconstruction problem can be solved using maximum-likelihood inference:

$$\underset{\boldsymbol{x} \in \mathbb{R}^{d \times d}}{\text{minimize}} \sum_{\psi \in \boldsymbol{\psi}} \|\mathcal{A}_\psi(\boldsymbol{x}) - \boldsymbol{y}_\psi\|_2^2. \tag{2}$$

A few important challenges arise. First, the CT problem is typically too high-dimensional to be solved in closed form, and hence one has to resort to iterative or gradient descent based schemes. This means that uncertainty quantification methods that require the second moment of the posterior distribution are not computationally feasible without further approximation. In particular, many active learning algorithms require an uncertainty estimate or samples from the posterior distribution, and are therefore challenging to implement in the CT setting. Second, in the sparse reconstruction regime ($n \times l < d^2$), the problem is underdetermined. The most common remedy is to add a regularizer (*e.g.*, L2 or Total-Variation loss), or other means of adding an inductive bias (*e.g.*, using non-linear representations and pre-training).

Turning now to the sequential experimental design setting, we consider the measurement space to be a uniform set of rotation angles $\Phi = \left\{i \cdot \Delta\phi\right\}_{i=0}^{N-1}$ out of which a subset $\boldsymbol{\psi}$ of size $n < N$ are going to be used for the reconstruction. In our setting, we use $N = 180$ angles with $\Delta\psi = 1°$ increment. The goal is to iteratively select those angles that are most informative in the sense that they yield the lowest reconstruction error. For selecting the angles, we proceed by sequentially selecting one angle at a time until the budget of $n$ angles is exhausted (see Algorithm 1).

## 3 DIFFUSION ACTIVE LEARNING

We now describe *Diffusion Active Learning* (DAL), a novel approach for data-driven, 'smart' angle selection in computed tomography. In a pre-training step, we train an unconditional Denoising Diffusion Probabilist Model (DDPM) on a training set consisting of tomographic slices from objects in

the domain of interest. This training is entirely independent of the inference (reconstruction) given the X-ray measurements, and requires only samples of tomographic slices to learn the image distribution. Our implementation uses the classical DDPM training (Ho et al., 2020). During the active learning loop, we use the trained diffusion model to approximate the posterior distribution conditioned on the current set of measurements (X-ray projections). We use a variant of the techniques proposed in Song et al. (2023) to generate these conditional samples. Given the relatively small size of the images in our study, we opt for a pixel-space diffusion model rather than a latent-space diffusion model. More generally, we emphasize that any generative posterior can be used for the active learning procedure described below. A more detailed comparison of different diffusion models is, however, beyond the scope of this work.

In the active learning loop, we use conditional samples from the diffusion model to approximate the uncertainty of the estimation. Using the forward model, we map the diffusion samples to the measurement space, to obtain the posterior distributions of the projections. Finally, we choose the angle that has the largest uncertainty to take the next measurement and repeat until our measurement budget is depleted. The active learning step is described in more detail in Section 3.2. The complete diffusion active learning framework is outlined in Algorithm 1.

From a Bayesian perspective, the Diffusion model corresponds to a learned prior $p(\boldsymbol{x})$ over images $\boldsymbol{x} \in \mathbb{R}^{d \times d}$. Diffusion posterior inference approximates the posterior, $p(\boldsymbol{x}|\boldsymbol{\mathcal{Y}}, \boldsymbol{\psi}) \propto p(\boldsymbol{x})p(\boldsymbol{\mathcal{Y}}|\boldsymbol{x}, \boldsymbol{\psi})$ for measurements $\boldsymbol{\mathcal{Y}}$. The likelihood term is specified by the forward model in Eq. (1) and the noise distribution. The posterior (mean) distribution of a new measurement $\boldsymbol{y}_{\psi_{new}} \in \mathbb{R}^d$ at angle $\psi_{new}$ is specified again by the forward model, marginalized over the posterior, $p(\boldsymbol{y}_{\psi_{new}}|\boldsymbol{\mathcal{Y}}, \boldsymbol{\psi}, \psi_{new}) = \int \mathcal{A}_{\psi_{new}}(\boldsymbol{x})p(\boldsymbol{x}|\boldsymbol{\mathcal{Y}}, \boldsymbol{\psi})d\boldsymbol{x}$. Equivalently, we can sample from the predictive posterior by sampling first from the posterior $\tilde{\boldsymbol{x}} \sim p(\boldsymbol{x}|\boldsymbol{\mathcal{Y}}, \boldsymbol{\psi})$ and applying the forward model to obtain $\tilde{\boldsymbol{y}}_{\psi_{new}} = \mathcal{A}_{\psi_{new}}(\tilde{\boldsymbol{x}})$. We choose the angle $\psi_{new}$ to maximize the total posterior variance, $\mathrm{tr}(\mathrm{Cov}[\tilde{\boldsymbol{y}}_{\psi_{new}}])$, also known as uncertainty sampling (Settles, 2009).

### 3.1 SCORE-BASED DIFFUSION MODELS AND CONDITIONAL SAMPLING

For a data distribution $p_0(\boldsymbol{x}_0) = p(\boldsymbol{x})$, a family of distributions $p_t(\boldsymbol{x}_t)$ can be defined by injecting i.i.d. Gaussian noise to data samples, such that $\boldsymbol{x}_t = \boldsymbol{x}_0 + \sigma_t\boldsymbol{\varepsilon}$ with $\boldsymbol{\varepsilon} \sim \mathcal{N}(0, I)$ and $\sigma_t$ monotonically increasing with respect to time $t \in [0, T]$. The score function $\nabla_{\boldsymbol{x}_t} \log p_t(\boldsymbol{x}_t)$ (i.e., gradient of log-probability) can be learned using a neural network via a denoising score matching objective $\mathcal{L}(\theta) = \mathbb{E}_{t, \boldsymbol{x}_0, \boldsymbol{x}_t}\left[\|\mathbf{s}_\theta(\boldsymbol{x}_t, t) - \nabla_{\boldsymbol{x}_t} \log p_t(\boldsymbol{x}_t|\boldsymbol{x}_0)\|_2^2\right]$ (Ho et al., 2020).

So far, the trained diffusion model is independent of measurements obtained during the active learning loop. At inference time, our objective is to sample from the posterior distribution by conditioning the diffusion model on X-ray projection measurements; this is done without any additional training. Our setup is similar to that of Chung et al. (2022) and Song et al. (2023). Given the set of current measurements $\boldsymbol{\mathcal{Y}}$ for angles $\boldsymbol{\psi}$, the goal is to sample from the posterior distribution $p(\boldsymbol{x}|\boldsymbol{\mathcal{Y}}, \boldsymbol{\psi})$. The conditional score at time $t$ can be obtained via Bayes' rule, where the second term still needs to be approximated $\nabla_{\boldsymbol{x}_t} \log p_t(\boldsymbol{x}_t|\boldsymbol{\mathcal{Y}}, \boldsymbol{\psi}) = \nabla_{\boldsymbol{x}_t} \log p_t(\boldsymbol{x}_t) + \nabla_{\boldsymbol{x}_t} \log p_t(\boldsymbol{\mathcal{Y}}, \boldsymbol{\psi}|\boldsymbol{x}_t)$. To this end, we use a variant of the *Hard Data Consistency* approach proposed by Song et al. (2023). At step $t$ of the reverse diffusion process, we start from our current noisy estimate $\boldsymbol{x}_t$, and use Tweedie's formula $\hat{\boldsymbol{x}}_0(\boldsymbol{x}_t) = \boldsymbol{x}_t + \sigma_t^2\mathbf{s}_\theta(\boldsymbol{x}_t, t)$ to get a noiseless estimate $\hat{\boldsymbol{x}}_0(\boldsymbol{x}_t)$. We then take several gradient steps solving the minimization problem (2) initialized with $\boldsymbol{x} = \hat{\boldsymbol{x}}_0(\boldsymbol{x}_t)$.

However, instead of fully solving the problem as done in Hard Data Consistency and then combining the result linearly with $\hat{\boldsymbol{x}}_0(\boldsymbol{x}_t)$, as proposed by Song et al. (2023) (for latent diffusion models), we use early stopping—that is, we perform a predefined, limited number of gradient steps to solve (2). We refer to this approach as *Soft Data Consistency*. Since the minimization is initialized with $\hat{\boldsymbol{x}}_0(\boldsymbol{x}_t)$, the resulting estimate $\boldsymbol{x}_0^*(\boldsymbol{x}_t)$ retains features of $\hat{\boldsymbol{x}}_0(\boldsymbol{x}_t)$, similar to the linear combination in Song et al. (2023), while promoting consistency with the current measurements. This approach avoids convergence to the exact solution, reducing computation time while maintaining solution quality. Moreover, since we use diffusion models in pixel space, we do not require complex scheduling to avoid the overhead of back-propagation through the latent diffusion model decoder, making our method simpler and faster for pixel-space diffusion models while achieving comparable performance to Hard Data Consistency.

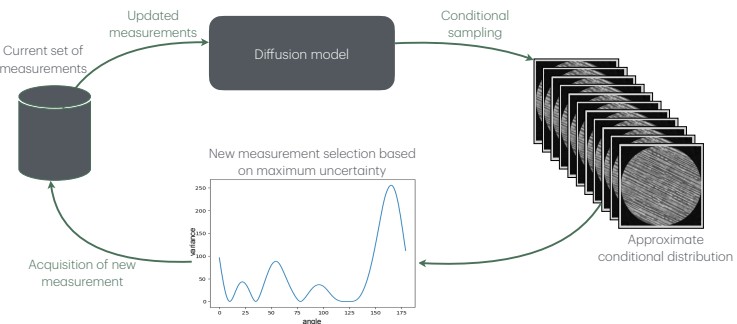

Figure 2: An illustration of the active learning loop using diffusion models to approximate the posterior distribution conditioned on current measurements.

Finally, $\boldsymbol{x}_0^*(\boldsymbol{x}_t)$ must be mapped back to the manifold defined by the noisy samples at time $t$, to go further the reverse diffusion process. To this end, we use the fact that $p(\boldsymbol{x}_t|\boldsymbol{x}_0^*(\boldsymbol{x}_t), \boldsymbol{\mathcal{Y}}, \boldsymbol{\psi})$ is a tractable Gaussian distribution whose mean is a scaling of $\boldsymbol{x}_0^*(\boldsymbol{x}_t)$. For further details and a comparison with Hard Data Consistency see Appendix A.

### 3.2 SAMPLING-BASED ACTIVE LEARNING

The selection process in active learning usually involves an information criteria (sometimes called acquisition function) which scores the informativeness of each possible measurement. To this end, most methods make use of uncertainty quantification, which, at the same time, poses one of the major challenges in the deep learning setting. The literature proposes a plethora of different acquisition functions (c.f. Settles, 2009; Gal et al., 2017; Ren et al., 2021), depending on the setting (*e.g.*, regression, classification), the statistical model (*e.g.*, linear, deep neural networks), and the learning target (*e.g.*, model identification, PAC). A common approach is to use a Bayesian model and (approximate) information theoretic measures such as mutual information. However, the pure Bayesian approach is often intractable beyond linear models, requiring further approximations.

We formulate the acquisition process in our sampling-based framework. At time step $t$ of the active learning loop, we sample $k$ images $\boldsymbol{x}_t^1, \ldots, \boldsymbol{x}_t^k \in \mathbb{R}^{d \times d}$ from the (approximate) posterior distribution. We use the conditional diffusion model to obtain the samples, but the formulation is general and works with any generative model. In this light, we can also view the sampled Laplace approximation of Barbano et al. (2022a) in the same framework. Intuitively, the samples $\boldsymbol{x}_t^1, \ldots, \boldsymbol{x}_t^k$ are consistent with the observation data, but differ in places where there is not enough data to constrain the posterior distribution. Our goal is to acquire additional measurements that differentiate the samples $\boldsymbol{x}_t^1, \ldots, \boldsymbol{x}_t^k$. We choose the angle $\psi_{t+1}$ that maximizes the posterior total variance,

$$\psi_{t+1} = \arg\max_{\psi \in \Phi} \frac{1}{k} \sum_{i=1}^{k} \|\mathcal{A}_\psi(\boldsymbol{x}_t^i) - \mathcal{A}_\psi(\bar{\boldsymbol{x}}_t)\|^2, \qquad \text{where} \quad \bar{\boldsymbol{x}}_t = \frac{1}{k} \sum_{i=1}^{k} \boldsymbol{x}_t^i. \qquad (3)$$

The optimization is over the discrete set of angles $\Phi$, therefore requires to apply the forward model $k \cdot |\Phi|$ times. In the tomographic setup, this computation can be batched efficiently on a GPU. Selecting the measurement angle that displays the largest sample variance introduces additional inference constraints in the consecutive round, in a way that effectively reduces the remaining variance in the posterior distribution. This is known as *uncertainty sampling* (Lewis, 1995) and has been analyzed formally in various settings (see, e.g. Settles, 2009; Liu & Li, 2023). We discuss alternative acquisition strategies in Appendix D, however we remark already now that we found no significant difference among the variants we tested. Therefore, based on our evaluation, we recommend Eq. (3) as a simple and yet effective choice.

## 4 EXPERIMENTS

In our experiments, we closely follow the setup introduced in Section 2. Diffusion active learning is implemented as described in Section 3. We evaluate the proposed approach and several baselines

Chip        Composite        Lung

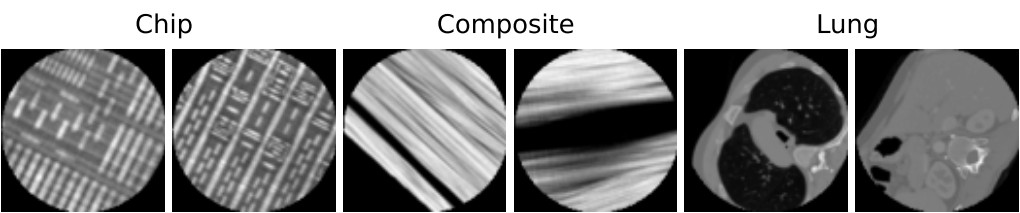

Figure 3: Two cropped and rescaled slices of size $128 \times 128$ from each of the test datasets.

on three real-world tomography image datasets as described below. The code is provided in the supplementary and will be released as open-source upon publication.

## 4.1 DATASETS

**Chip Data.** Our first dataset is an integrated circuit measured with ptychographic X-ray laminography (PyXL) (Holler et al., 2019b;a). PyXL uses ptychography to scan the sample with a coherent X-ray beam, acquiring diffraction patterns for projection reconstruction at multiple rotation angles. The sample rotation axis is tilted relative to the detector, allowing high-resolution, non-destructive imaging of planar samples across various scales. The dataset features a 3D volume of an integrated circuit, with large metal interconnects in the upper layers and progressively smaller features towards the transistor layer at the bottom.

**Composite Materials.** As a second dataset, we use tomography data of a composite material (Auenhammer et al., 2020a;b). This dataset contains 3D tomographic reconstructions of non-crimp fabric reinforced composites, which captures the arrangement and orientation of the fiber bundles and the matrix in which they are embedded. Such composites are extensively used in wind turbine blades and consist of fiber bundles aligned in one direction with stitching yarns to aid in manufacturing and handling. Hence, this dataset was used for the creation of precise finite element models to simulate and analyze the mechanical behavior of the materials, particularly their stiffness and response to fatigue.

**Lung Data.** The LIDC/IDRI dataset (Armato III et al., 2011) is designed specifically for training and comparing deep learning-based methods for low-dose CT reconstruction, and consists of helical thoracic CT scans. We chose 40,000 CT scan slices from LIDC/IDRI , data from approximately 800 patients, to define the dataset used in this paper. We use the same 40,000 examples chosen by Leuschner et al. (2019).

Due to computational constraints, we worked with two image sizes namely, $128 \times 128$ and $512 \times 512$ pixels. To produce smaller images we used cropping and rescaling of the slices of the reconstructed 3D objects in the datasets. For $128 \times 128$ images, we took $256 \times 256$ crops from the chip and composite material slices, and then rescaled them to size $128 \times 128$ using bilinear interpolation. For the Lung dataset, we rescaled directly each of the slices in the dataset to $128 \times 128$. For the $512 \times 512$ images, we simply took crops of size $512 \times 512$ from the CT reconstruction datasets described above. All these crops are independent with no overlap. The final images are then split into disjoint train and test datasets.

In all cases, we use the Radon transform to generate the projection data synthetically using parallel beam geometry. For training the diffusion models, we use data augmentation on the train data by random rotation and scaling (1x to 1.3x). The evaluation on the test set was run on P100 compute nodes (one node per instance). For pre-training the diffusion models, we used a single A100 node.

## 4.2 METHODS

**SWAG.** Maddox et al. (2019) propose SWA-Gaussian (SWAG), a simple method to enhance uncertainty representation and calibration in deep learning models. SWAG extends Stochastic Weight Averaging (SWA) by capturing the first moment (mean) of the weights using SWA (Izmailov et al.,

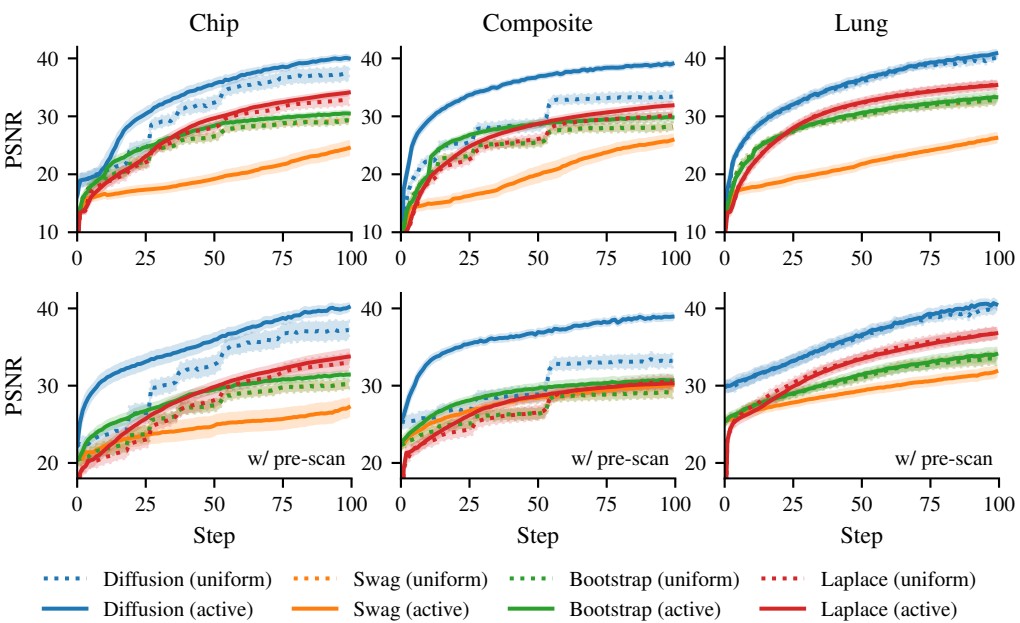

Figure 4: Benchmark for $128 \times 128$ images, averaged over 30 data points. Confidence bands show two times standard error. Top (bottom) row correspond to results without (with) low-res pre-scan.

2018), and then modeling the second moment (covariance) using a low-rank plus diagonal approach. This forms an approximate Bayesian posterior distribution over the neural network weights. The approach utilizes the stochastic gradient descent (SGD) trajectory to estimate the posterior's mean and covariance efficiently, leveraging the observed behavior that SGD iterates approximate a Gaussian distribution in the parameter space of deep networks. As our active learning approach is itself based on samples from the posterior distribution, we directly subsample the SGD trajectory around the mode instead of fitting a Gaussian distribution and resampling, leading to no significant differences in the evaluation.

**Bootstrap.** A second way of using the inherent randomness of the SGD trajectory is to train multiple instances of the model. Lakshminarayanan et al. (2017) present a scalable method for predictive uncertainty estimation in deep neural networks, referred to as *Deep Ensembles* (or *Bootstrap*). This approach avoids the complexities and computational burdens of Bayesian neural networks by instead employing ensembles of neural networks to approximate uncertainty. Each network in the ensemble is trained independently on the same dataset, leveraging random initializations to induce diversity among the models. Predictive uncertainty is then quantified by the empirical distribution of the ensemble members.

**Laplace Approximation and Deep Image Prior.** While closed-form solutions of the linear model Eq. (2) are computationally intractable due to the high-dimensional input and observation spaces, Antoran et al. (2023) have proposed a kernelized and optimize-based approximation applicable specifically in the CT setting. This approach was used for Bayesian experimental design (Barbano et al., 2022a) in combination with a linearized deep image prior (DIP) network (Ulyanov et al., 2018). The method introduces a Gaussian surrogate for the total variation (TV) regularizer to preserve Gaussian-linear conjugacy, enabling efficient computation of uncertainty measures. However, the experimental evaluation of Barbano et al. (2022a) is still restricted to a synthetic toy example. Expanding on this framework, Antoran et al. (2022) introduce a scalable sampling-based approach to uncertainty quantification in large-scale linear models. They extend this approach to non-linear parameteric models using a linearized Laplace approximation. To calibrate the posterior hyperparameters, they propose an efficient Expectation Maximization scheme for marginal likelihood optimization. Our evaluation uses the reference implementation of Antoran et al. (2023).

| Dataset | Diffusion | Swag | Bootstrap | Laplace |
|---|---|---|---|---|
| Chip | **27** $[-2, +3]$ | $> 100$ | $80 [-8, +14]$ | $53 [-4, +4]$ |
| Composite | **15** $[-3, +1]$ | $> 100$ | $> 100$ | $65 [-6, +8]$ |
| Lung | **18** $[-2, +3]$ | $> 100$ | $44 [-4, +7]$ | $34 [-3, +4]$ |

Table 1: Number of measurements required to achieve an average score of PSNR of 30 dB using active learning acquisition for the different models and datasets studied in this paper. Ranges ares computed using two times standard error. Diffusion active learning achieves up to a $4.3\times$ improvement compared to the Laplace model on the Composite data.

## 4.3 RESULTS

We compared the performance of several baselines including four different generative models and non-adaptive uniform acquisition (computed sequentially by halving the remaining angle space, *e.g.*, $0°, 90°, 45°, 135°, \dots$). As generative models, we consider diffusion models, SWAG, Bootstrap, and Laplace, and for acquisition we consider a non-adaptive, uniform allocation, and active learning based on (3). We consider other acquisition function in Appendix D, but we found no significant difference among the tested variants. In addition, we consider two settings: with and without *pre-scan*. In the pre-scan setting, there is a previously computed low-resolution scan of the object that can be used as a prior in the reconstruction process. Pre-scanning is a common procedure at synchrotron X-ray beamlines to get a quick estimate of the sample using much lower radiation. All models are conditioned on the pre-scan data in the same way as for the other measurements. Our experiments show that diffusion models with active learning consistently outperforms other methods in terms of Peak Signal-to-Noise Ratio (PSNR) and computational efficiency. We showcase other metrics in Appendix C, which also follow a similar pattern.

Figure 4 summarizes the performance on our three datasets for images of size $128 \times 128$. If no pre-scan is used, diffusion models outperform all other generative models in terms of PSNR with up to $4.3\times$ reduction in the number of measurements needed and, while the gap is dataset-dependent, there is always a clear advantage in using diffusion models to generate tomographic reconstructions. In terms of acquisition functions, we benchmark against uniform acquisition in angular space. While being data-distribution independent, uniform remains on par with active learning acquisition functions for the Lung dataset. This can be attributed to the fact that there are not clear directions of preference for lung images and their features appear isotropic in all directions. For chip and composite materials samples however, active learning acquisitions based in (3) are able to identify the most relevant directions, and clearly outperform the uniform strategy. Table 1 summarizes the gains of diffusion active learning in terms of number of measurements for a target PSNR of 30 dB.

When using pre-scan, the different generative models can make out the rough shape of the object from the beginning of the experiment, which leads into higher quality reconstructions over the whole range of measurements. This in turn provides more information to decide which measurement to take next. This is particularly noticeable for SWAG which becomes competitive with Bootstrap and Laplace. Diffusion models still provide better reconstructions. However, after sufficient number of measurements (*e.g.*, 50 for Lung data and 100 for chip data) there is no clear advantage in using the learned prior.

Figure 5 corroborates our findings with larger images of $512 \times 512$ pixels tested on the chip dataset. The results are consistent with the $128 \times 128$ case, showcasing that the advantages in the use of both diffusion models and active learning acquisitions are not restricted to low resolution scans. Figure 5 further shows computation times for the different methods. At $512 \times 512$ resolution, DAL requires less than two minutes per step; significantly faster than the Laplace approach by Barbano et al. (2022a). In the context of long image acquisition times of up to several days (Aidukas et al., 2024), the proposed method can provide a significant advantage by reducing the data requirements.

## 5 CONCLUSION

We introduced *Diffusion Active Learning* (DAL), a novel framework that combines generative diffusion models with active learning for CT reconstruction. Our experimental evaluation showcases

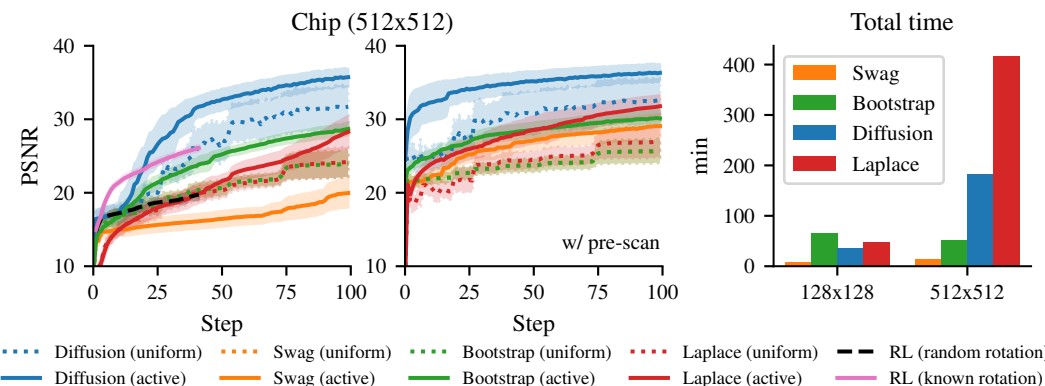

Figure 5: Left: Benchmark for $512 \times 512$ images averaged over 10 data points. Confidence bands show two times standard error. Right: Average running time for 100 steps of active learning using different models.

significant gains in improving the reconstruction quality with fewer samples compared to static uniform allocation and several baselines from prior works. The reduction in X-ray dose and cost savings achieved by diffusion active learning have significant implications for practical CT applications. These improvements can lead to wider adoption in scientific imaging and material sciences at synchrotron facilities, due to shorter X-ray beam-time allocation, faster experiments and enhanced quality. As noticed in our results, the gains provided by our method are dataset dependent, showing improvements in particular with highly structured images. Lastly, we remark that the DAL framework applies to any (differentiable) forward process, and as such can be applied to other setups such as MRI or ptychographic reconstruction methods.

As often, the achieved gains come with trade-offs. First, training the diffusion model requires a sufficient training data in the domain of interest. This is a reasonable assumption for many applications where prior reconstruction data is readily available. A possible way forward is to pre-train large foundation models on a variety of tomographic images, or even use pre-trained models such as Stable Diffusion (Podell et al., 2023) for fine-tuning on much smaller datasets (Hu et al., 2021).

Second, while diffusion models are computationally more costly than iterative reconstruction or filtered back-projection, our methods run on an NVIDIA P100 with 16GB of memory for resolutions up to 1024x1024, displaying reasonable scaling properties with many possibilities for further improvements. In addition, image acquisition time in micro- and nano-tomography can take up to several days (Aidukas et al., 2024), providing a clear use-case where the additional computational effort can be justified for improving sample efficiency.

Lastly, the key advantage of a learned prior comes at the cost of introducing reconstruction bias in cases where the measured sample is not contained in the training distribution; this is in particular relevant when the goal is to detect small deviations or defects in the samples. The same reason might prohibit the use of the proposed approach in high-stakes setting such as medical applications, or at least not without enhanced safety measures. The work of Barbano et al. (2023) takes a first steps towards working with out-of-distribution samples. More generally, in a real-world tomography experimental setup, many additional challenges arise, including sample alignment and measurement noise. Despite our effort of making the evaluation more realistic, there remains a sim-to-real gap to be addressed in future works, *e.g.*, by considering distribution shifts in the testing distribution.

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

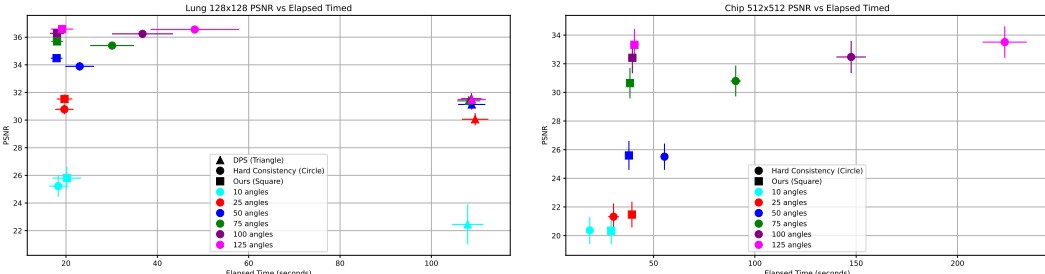

Figure 6: Ablation of soft (Square) vs hard data consistency (disk) on both Lung and Chip datasets. We evaluate the performance with different sparsity. While the quality is comparable, our approach requires a fraction of the time. We use SGD consistency with a fixed number of steps, each with a fixed batch size, our running time is then predictable and stable, without any loss in quality.

## A BACKGROUND ON SCORE-BASED DIFFUSION MODELS AND CONDITIONAL SAMPLING

We succinctly review the fundamentals of diffusion models, namely the formulation of denoising diffusion probabilistic model (DDPM) (Ho et al., 2020). The forward diffusion process incrementally adds Gaussian noise to the data. Let $\boldsymbol{x}_0 \sim p(\boldsymbol{x})$ denote the initial data sample, and $\boldsymbol{x}_t$ the data at time step $t$. The forward noising process can be described by the following stochastic differential equation (SDE):

$$d\boldsymbol{x}_t = -\frac{\beta_t}{2}\boldsymbol{x}_t dt + \sqrt{\beta_t}d\boldsymbol{w}\,, \tag{4}$$

where $\beta_t$ is the noise scheduler, and $\boldsymbol{w}$ represents the standard Wiener process.

The reverse process aims to recover the original data from the noised version by reversing the SDE:

$$d\boldsymbol{x}_t = \left[\frac{\beta_t}{2}\boldsymbol{x}_t - \beta_t \nabla_{\boldsymbol{x}_t} \log p_t(\boldsymbol{x}_t)\right]dt + \sqrt{\beta_t}d\boldsymbol{w}_t, \,. \tag{5}$$

Here, $\nabla_{\boldsymbol{x}_t} \log p_t(\boldsymbol{x}_t)$ is the score function, which is the gradient of the log probability density at time $t$. We train a diffusion model $\mathbf{s}_\theta(x_t, t)$ to approximate the true score function. The training objective is typically:

$$\mathcal{L}(\theta) = \mathbb{E}_{t,\boldsymbol{x}_0,\boldsymbol{x}_t}\left[\|\mathbf{s}_\theta(\boldsymbol{x}_t, t) - \nabla_{\boldsymbol{x}_t} \log p_t(\boldsymbol{x}_t|\boldsymbol{x}_0)\|_2^2\right]. \tag{6}$$

Once the score function is learned, data samples can be generated by solving the reverse SDE using numerical methods such as Euler-Maruyama or more sophisticated solvers.

In order to improve the sampling speed of DDPM, Song et al. (2020) proposed the denoising diffusion implicit model (DDIM) which defines the diffusion process as a non-Markovian process. A crucial step to achieve this is to notice that one can predict a noiseless variant of $\hat{\boldsymbol{x}}_0(\boldsymbol{x}_t)$ from $\boldsymbol{x}$ using Tweedie's formula

$$\hat{\boldsymbol{x}}_0(\boldsymbol{x}_t) = \frac{1}{\overline{\alpha}_t}\big(\boldsymbol{x}_t + \sqrt{1 - \overline{\alpha}_t}\mathbf{s}_\theta(\boldsymbol{x}_t, t)\big), \tag{7}$$

where $\alpha_t = 1 - \beta_t$ and $\overline{\alpha}_t = \Pi_{i=1}^t \alpha_i$.

**Solving Inverse Problems using Score-Based Diffusion Models.** Scientific inverse problems like CT, where we only have access to partial information due to the limited number of observations, are inherently ill-posed and hence, no unique reconstruction of $\boldsymbol{x}$ is possible.

To address this problem, we learn a prior $p(\boldsymbol{x})$ from the training set, and sample from the posterior distribution $p_t(\boldsymbol{x}|\boldsymbol{\mathcal{Y}}, \boldsymbol{\psi})$. To achieve this goal, Chung et al. (2022) introduced Diffusion Posterior Sampling. Using a pre-trained score-based model as a prior, they modified (5) and obtain a reverse diffusion process to sample from the posterior distribution:

$$d\boldsymbol{x}_t = \left[\frac{\beta_t}{2}\boldsymbol{x}_t - \beta_t(\nabla_{\boldsymbol{x}_t} \log p_t(\boldsymbol{x}_t) + \nabla_{\boldsymbol{x}_t} \log p_t(\boldsymbol{\mathcal{Y}}, \boldsymbol{\psi}|\boldsymbol{x}_t))\right]dt + \sqrt{\beta_t}d\boldsymbol{w}_t, \tag{8}$$

where they use the fact that

$$\nabla_{\boldsymbol{x}_t} \log p_t(\boldsymbol{x}_t | \boldsymbol{\mathcal{Y}}, \boldsymbol{\psi}) = \nabla_{\boldsymbol{x}_t} \log p_t(\boldsymbol{x}_t) + \nabla_{\boldsymbol{x}_t} \log p_t(\boldsymbol{\mathcal{Y}}, \boldsymbol{\psi} | \boldsymbol{x}_t). \tag{9}$$

Using (7) allows to use the forward operator of the inverse problem on $\hat{\boldsymbol{x}}_0(\boldsymbol{x}_t)$ to obtain an estimate $\hat{\boldsymbol{\mathcal{Y}}}_t$ of the measurements. By defining a loss between $\hat{\boldsymbol{\mathcal{Y}}}_t$ and the true measurements $\boldsymbol{\mathcal{Y}}$, we can back-propagate the error to $\boldsymbol{x}_t$ and obtain an approximation of $\nabla_{\boldsymbol{x}_t} \log p_t(\boldsymbol{\mathcal{Y}}, \boldsymbol{\psi} | \boldsymbol{x}_t)$.

Based on these ideas, Song et al. (2023) proposed the use latent diffusion models, instead of pixel-space diffusion models, and introduced a variant of the conditional sampling. Additionally, they proposed Hard Data Consistency, which consists in using (7) to obtain an estimate $\hat{\boldsymbol{x}}_0(\boldsymbol{x}_t)$ and then solve completely the following optimization problem initialized with $\hat{\boldsymbol{x}}_0(\boldsymbol{x}_t)$:

$$\boldsymbol{x}_0^*(\boldsymbol{\mathcal{Y}}, \boldsymbol{\psi}) \in \underset{\boldsymbol{x} \in \mathbb{R}^{d \times d}}{\arg\min} \sum_{\psi \in \boldsymbol{\psi}} \|\mathcal{A}_\psi(\boldsymbol{x}) - \boldsymbol{y}_\psi\|_2^2 \tag{10}$$

where $\boldsymbol{\psi}$ is the current set of angles measured, and $\mathcal{A}_\psi$ is the Radon transform with angle $\psi$. Notice that if the inverse problem is ill-posed, the initialization and the optimization algorithm determine the value of $\boldsymbol{x}_0^*(\boldsymbol{\mathcal{Y}}, \boldsymbol{\psi})$. Finally, one needs to map $\boldsymbol{x}_0^*(\boldsymbol{\mathcal{Y}}, \boldsymbol{\psi})$ back to the manifold defined by the noisy samples at time $t$, to go further with the reverse diffusion process. To this end, Song et al. (2023) proposed *Stochastic Encoding*, which uses the fact that $p(\boldsymbol{x}_t | \boldsymbol{x}_0^*(\boldsymbol{\mathcal{Y}}, \boldsymbol{\psi}), \boldsymbol{\mathcal{Y}}, \boldsymbol{\psi})$ is a tractable Gaussian distribution with mean being a scaling of $\boldsymbol{x}_0^*(\boldsymbol{\mathcal{Y}}, \boldsymbol{\psi})$. However, since the measurements might be noisy, and since we have an ill-posed problem, using solely $\boldsymbol{x}_0^*(\boldsymbol{\mathcal{Y}}, \boldsymbol{\psi})$ leads to noisy image reconstructions. Thus, they propose a variance reduction by taking a linear combination between $\boldsymbol{x}_0^*(\boldsymbol{\mathcal{Y}}, \boldsymbol{\psi})$ and $\hat{\boldsymbol{x}}_0(\boldsymbol{x}_t)$ before mapping it back, a method which they coined *ReSample*.

**Soft Data Consistency and Early Stopping.** Our setup is similar to that of Chung et al. (2022) and Song et al. (2023). We first pre-train a diffusion model on domain-specific tomogram data that captures the underlying distribution of the desired application. We then condition on the set of current measurements, and produce samples of the posterior distribution using an approach similar to that of *Hard Data Consistency* (Song et al., 2023). More specifically, at step $t$ of the reverse diffusion process, we start from our current estimate $\boldsymbol{x}_t$, and use Tweedie's formula (7) to get a noiseless estimate $\hat{\boldsymbol{x}}_0(\boldsymbol{x}_t)$. We then take several *consistency* gradient steps solving the following minimization problem initialized with $\boldsymbol{x} = \hat{\boldsymbol{x}}_0(\boldsymbol{x}_t)$,

$$\underset{\boldsymbol{x}}{\text{minimize}} \sum_{\psi \in \boldsymbol{\psi}} \|\mathcal{A}_\psi(\boldsymbol{x}) - \boldsymbol{y}_\psi\|_2^2. \tag{11}$$

However, instead of solving the problem completely as in Hard Data Consistency, and then taking a linear combination with $\hat{\boldsymbol{x}}_0(\boldsymbol{x}_t)$ as done in ReSample proposed by Song et al. (2023), we instead take a few steps of Stochastic Gradient Descent (SGD), effectively applying early stopping. Since the minimization problem is initialized with $\hat{\boldsymbol{x}}_0(\boldsymbol{x}_t)$, this effectively ends with an estimate that retains features of $\hat{\boldsymbol{x}}_0(\boldsymbol{x}_t)$ while encouraging consistency with the current measurements. Moreover, it reduces the computational cost and allows us to use data consistency in each step of the reversed diffusion process. This is not the case with ReSample, where they have to pick a small subset of time steps in which to perform the consistency optimization. We finally use Stochastic Encoding to map back our estimate to the manifold defined by the noisy samples at time $t$ and continue with the reversed diffusion process.

Figure 6 shows a comparison of our method with Hard Data Consistency (Song et al., 2023) and Diffusion Posterior Sampling Song et al. (2023). Since we use SGD for consistency steps with a fixed batch-size, the running time of our approach is the same regardless of the number of angles sampled. This is not the case for Hard Data Consistency, where they implement full batch updates in each update, and where they run until a certain specified threshold has been reached with the loss in equation ( 11). This takes longer and longer as more angles are sampled. Moreover, this training has higher variance as can be seen in Figure 6. In contrast, we have a fixed number of consistency gradient steps after each diffusion step, hence the running time of our algorithm is predictable and constant.

In their implementation, Song et al. (2023) notices already that they could not apply hard consistency over the entire diffusion process. They therefore partitioned the time in 3 equal sections. In the first,

they perform no conditioning, in the second, they perform the same update that we do; a fixed number of gradient steps per diffusion step. Finally, it is only in their third phase that they introduce the hard consistency. This is how we implemented it for our comparison. While they implemented it for latent diffusion models, some of their ideas can be repurposed to work with pixel-space diffusion models, but they needed a strong adaptation that we presented in this paper.

## B  DIFFUSION ACTIVE LEARNING FOR MRI

MRI operates within the same framework described in Equation (1), with $\boldsymbol{y}_{\psi} = \mathcal{A}_{\psi}(\boldsymbol{x}) + \epsilon$, where $x$ is a complex value object, $\mathcal{A}_{\psi}(\boldsymbol{x})$ is a forward model consisting of taking the Fourier transform, which produces the *k-space* of $\boldsymbol{x}$, followed by a masking that takes a single row (or column) specified by the index $\psi$. The set of possible measurements is then the set $\boldsymbol{\mathcal{Y}}_{\psi}$ of all possible rows of the k-space of $\boldsymbol{x}$. Assuming a Gaussian distribution of the noise $\epsilon$, the reconstruction problem can be solved using maximum-likelihood inference given by Equation 2.

Given the similarity between this setup and the CT setup studied in the main body of this paper, we extend our framework to work with MRI reconstructions using the formulation of Diffusion Active Learning and Algorithm 1.

**Datasets.**  We benchmarked our algorithm with the FastMRI knee dataset (Zbontar et al., 2018), which is a large-scale open dataset designed to accelerate research in magnetic resonance imaging (MRI) reconstruction using machine learning. It consists of raw k-space data and fully-sampled ground truth images, enabling both supervised and unsupervised training. We focus on knee scans acquired using single coils. We trained our diffusion model with the provided train dataset and tested it on the disjoint test set.

**Methods.**  We compare against the same methods introduced in the main body of the paper for the CT setting, i.e., SWAG and Bootstrap. Laplace is not included as we were relying on the author's implementation which does not support the MRI model at this time. Each of the baselines works with active acquisition using uncertainty sampling (maximizing the 'variance' score), and a non-adaptive baseline that selects from a uniform pool of angles ('uniform') or columns corresponding to frequencies in increasing order ('low to high').

**Results.**  Figure 7 summarizes our results for the FastMRI dataset. As generative models, we consider diffusion models, SWAG, and Bootstrap; and for acquisition, we consider two non-adaptive allocations , *uniform* and *low to high*, explained below, and the active learning allocation based on Equation (3). On each acquisition step, both non-adaptive allocations choose the non-measured row closest to the center from a pool of predefined available rows. For an AL process with $k$ steps, the pool of rows for the uniform allocation consists of $k$ equispaced rows, while for the low to high allocation, the pool consists of the $k$ rows closest to the center, i.e., the ones with the lowest frequencies.

As in the case of CT, the diffusion model outperforms all other generative models in terms of PSNR with up to $2\times$ reduction in the number of sampled rows needed to achieve the same PSNR.

In MRI, similar to the pre-scan in CT, it is common to pre-scan the first few rows with lower frequencies. We examine two cases: 2 and 30 rows preselected closest to the center. As anticipated, the preselected rows with lower frequencies enable the generative model to obtain more accurate predictions through various sparsity levels. While diffusion models retain the lead, the other models catch up more quickly.

While we believe the performance of Active Learning strategies is also data-distribution dependent, we can see that for FastMRI, DAL clearly outperforms the non-adaptive allocation, which was not the case in CT medical images of the Lung dataset, though the gap is narrow with low-to-high allocations. The uniform allocation, however, struggles to generate meaningful reconstructions with only two preselected rows, likely meaning that some important low frequencies were skipped. For 30 preselected rows, uniform is more competitive, but is eventually clearly outperformed.

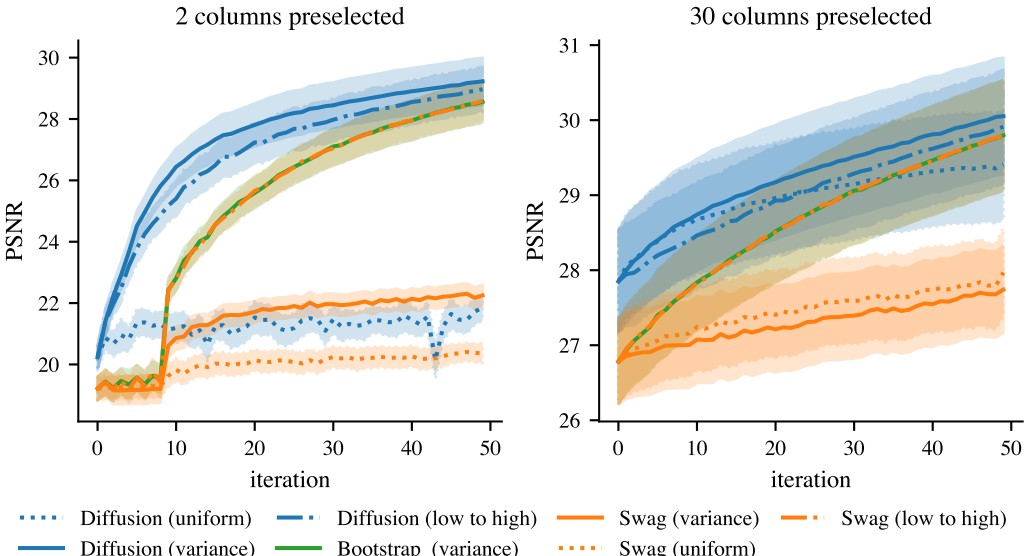

Figure 7: Benchmark over 26 test images of the FastMRI dataset. Confidence bands show two times standard error. The reconstructions of Bootstrap and Swag are equivalent to solving the inverse Fourier problem given the observed columns in k-space; for active learning selection ('variance), a separate copy of the model is used for random sampling. Sampling 50 rows corresponds to an acceleration of $6.38\times$ for 2 columns and $4.1\times$ for 30 columns.

Fig. 17 and Fig. 18 show qualitative results of the reconstructions. Here we can see that the diffusion model obtains a sharper image with as few as 10 adaptively chosen measurements (plus two pre-selected columns), while the other generative models struggle to obtain a meaningful reconstruction.

**Accelerated MRI diffusion inference.** While diffusion inference can be done by optimizing Equation (2) as in the CT case using gradient descent steps, we developed an accelerated formulation for MRI described as follows. We recall first the optimization objective of Equation (2):

$$\underset{\boldsymbol{x}\in\mathbb{R}^{d\times d}}{\text{minimize}} \sum_{\psi\in\boldsymbol{\psi}} \|\mathcal{A}_\psi(\boldsymbol{x}) - \boldsymbol{y}_\psi\|_2^2.$$

Notice that a solution to the optimization is given by computing the k-space of $\boldsymbol{x}$ and then replacing all the rows in $\boldsymbol{\mathcal{Y}_\psi}$ by the measurements $\{\boldsymbol{y}_\psi|\psi \in \boldsymbol{\mathcal{Y}_\psi}\}$. That is, we perform a *Fourier in-painting*, after which we apply the inverse Fourier transform to go back to pixel space and obtain our estimate $\hat{\boldsymbol{x}}$ being a minimizer of Equation (2). As shown in Figure 8, using this formulation leads to an up to $4\times$ improvement in inference time, and hence, also similar gains in the entire loop of DAL.

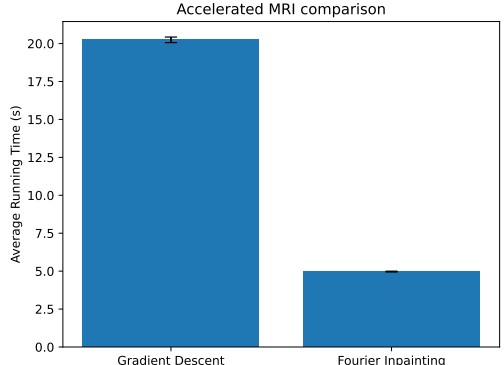

Figure 8: Compares the average running time of diffusion MRI inferences: gradient descent updates vs Fourier in-painting. Fourier in-painting achieves a $4\times$ acceleration. Results are averaged over 30 independent inferences, and bars show the standard error.

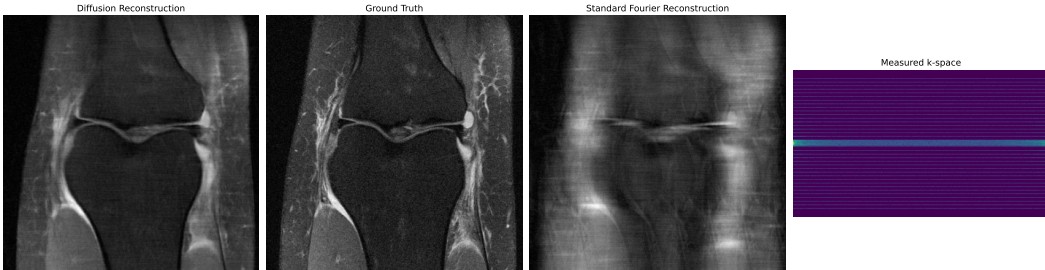

Figure 9: A diffusion reconstruction for a sample of the FastMRI dataset and its comparison with the standard Fourier reconstruction.

## C  EXPERIMENTS: ADDITIONAL DETAILS

### C.1  COMPUTING PARAMETERS

To sample from our diffusion models, we use 50 steps of the reverse diffusion process out of 1000 using the DDIM scheduler. For the soft data consistency, we use 50 gradient steps of the loss in equation (2). A single reconstruction in an A100 chip takes around 10 seconds, while 10 reconstructions take up to 25 seconds using less than 16GB of memory for images of size 512x512. Figure 5 (right) further shows the average running time of an entire active learning loop with 100 iterations for image sizes 512x512 and 128x128 on a P100 GPU.

### C.2  DATASET

A visualization of the three datasets introduced in Section 4 is given in Fig. 10 below.

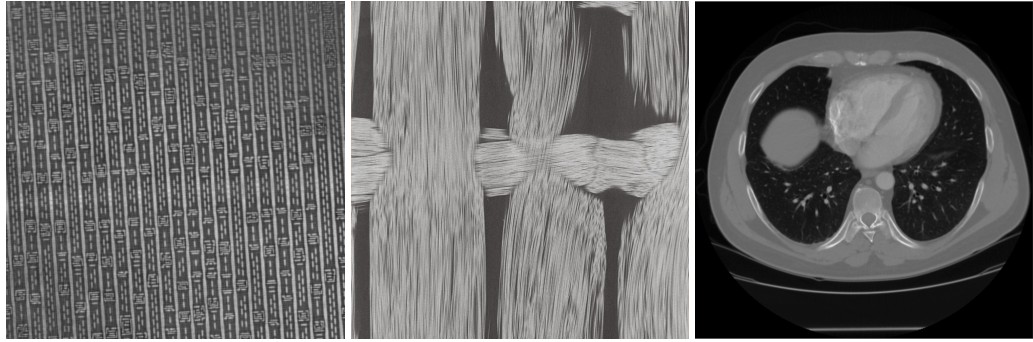

Figure 10: A 1000x1000 pixels crop sample of each of our three datasets, each before rescaling.

### C.3  ABLATION: RECONSTRUCTION METHOD AND ANGLE SELECTION

An interesting question is if the advantage of diffusion active learning is due to better angle selection or due to better reconstruction, or a combination of both. To answer this question, we re-evaluated the sequence of angles selected by the Bootstrap model using uncertainty sampling, and performed the reconstruction using the diffusion model. As expected, this improves the PSNR of the reconstruction. However, the resulting image quality as measured by PSNR is still significantly below the quality achieved by diffusion active learning. This showcases that the best reconstruction is achieved by the combination of both: The diffusion model captures the data distribution, and the angles selected by diffusion active learning exploit the data distribution in a way that a distribution-independent approach cannot; Bootstrap uses only information obtained from the current sample, and therefore intuitively cannot "reason" about the posterior distribution as the diffusion model

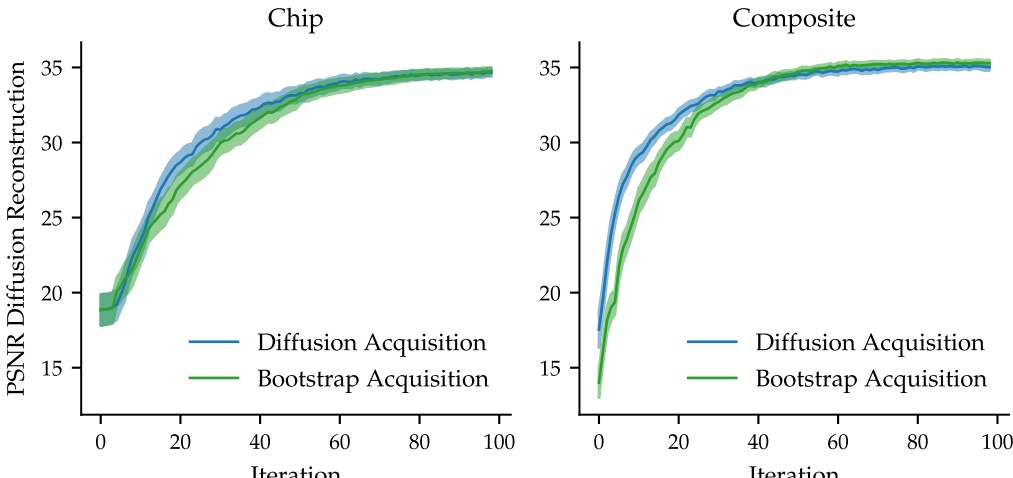

Figure 11: The plots show PSNR using Diffusion Posterior Sampling as a reconstruction method (using soft data consistency), with a sequence of angles selected either by Diffusion Active Learning or the Bootstrap uncertainty sampling. While eventually, both diffusion reconstructions reach the same PSNR score, for fewer angles there is a clear advantage of using Diffusion Active Learning for angle selection. This showcases that the selected sequence of angles is specific to the reconstruction method. In particular, the learned diffusion prior is not only used for better reconstruction, but is also used to choose a sequence of measurements that leads to better reconstructions.

does. Note also that the gains are not purely from better angle selection, as there is a significant gap between uniform and active selection on the composite and chip data.

## C.4 ADDITIONAL EVALUATION METRICS

We provide plots for PSNR, RMSE and SSIM metrics in Fig. 12 below. Note that the trend for all metrics is the same, showing that active learning significantly outperforms uniform acquisition for the Chip and Composite data, and achieving similar construction quality for the Lung data set.

## C.5 VISUALIZATION OF RECONSTRUCTIONS

To allow for a qualitative comparison, we provide the reconstructed images at steps 1,10,20 and 30 below in Figs. 13 to 16, for the active learning acquisition and without pre-scan. Note that visually the reconstruction quality of the diffusion model is already superior after 10 steps, and displaying intricate details of the $512 \times 512$ reconstruction with as few as 20 projections.

## D SAMPLING-BASED ACTIVE LEARNING

### D.1 ACQUISITION FUNCTIONS

We provide further details on the acquisition process of active learning, and discuss several acquisition function proposed in the literature (e.g., Settles, 2009) that can be applied in the tomographic reconstruction setting.

**Entropy and Mutual Information** Assume, for a moment, that at step $t$ of the acquisition process with observations $\boldsymbol{\mathcal{Y}}_t$ for angles $\boldsymbol{\psi}_t$, we have an exact and fully specified Bayesian model with a posterior over images, $p_t(\boldsymbol{x}) := p(\boldsymbol{x}|\boldsymbol{\mathcal{Y}}_t, \boldsymbol{\psi}_t) \propto p(\boldsymbol{x})p(\boldsymbol{\mathcal{Y}}|\boldsymbol{x}_t, \boldsymbol{\psi}_t)$. The observation likelihood $p(\boldsymbol{\mathcal{Y}}|\boldsymbol{x}, \psi)$ is defined by Eq. (1) for taking a new measurement at angle $\psi$. As our goal is to reconstruct the image $\boldsymbol{x}$, a natural acquisition target is the conditional mutual information between the

reconstruction $\boldsymbol{x}$ and the observation conditioned on the choice of angle $\psi_{t+1} = \psi$,

$$\psi_{t+1} = \arg\max_{\psi \in \Phi} \mathbb{I}_t(\boldsymbol{x}; \boldsymbol{\mathcal{Y}}_{t+1}|\psi_{t+1} = \psi). \tag{12}$$

The subscript $t$ indicates conditioning on the observed data filtration, $\mathbb{I}_t(\boldsymbol{x}; \boldsymbol{\mathcal{Y}}_{t+1}|\psi_{t+1} = \psi) :=$ $\mathbb{I}(\boldsymbol{x}; \boldsymbol{\mathcal{Y}}_{t+1}|\psi_{t+1} = \psi, \boldsymbol{\mathcal{Y}}_t, \boldsymbol{\psi}_t)$. Rewriting the mutual information using the entropy,

$$\mathbb{I}_t(\boldsymbol{x}; \boldsymbol{\mathcal{Y}}_{t+1}|\psi_{t+1} = \psi) = \mathbb{H}_t(\boldsymbol{\mathcal{Y}}_{t+1}|\psi_{t+1} = \psi) - \mathbb{H}_t(\boldsymbol{\mathcal{Y}}_{t+1}|\boldsymbol{x}, \psi_{t+1} = \psi) \tag{13}$$

From this, we note that maximizing the mutual information Eq. (12) is the same as maximizing the posterior entropy of the observation in cases where the observation distribution is independent of the angle (e.g. homoscedastic Gaussian noise models). In practice, computing the mutual information or the entropy is computationally challenging except for in special cases such as a Gaussian conjugate model. Assuming that the posterior distribution is $\mathcal{N}(\boldsymbol{x}_t, \Sigma_t)$ with covariance $\Sigma_t \in \mathbb{R}^{(d \times d)^2}$ and the likelihood is also Gaussian, centered at $\mathcal{A}_\psi(\boldsymbol{x}_t)$ with variance $\sigma$, i.e. $\mathcal{N}(\boldsymbol{x}_t, \sigma\boldsymbol{1}_l)$ with the unit matrix $\boldsymbol{1}_l \in \mathbb{R}^{l \times l}$, the close form of Eq. (12) is

$$\log\det(\sigma\boldsymbol{1}_l + A_\psi \Sigma_t A_\psi^\top) + C, \tag{14}$$

where $C$ is a constant that does not depend on the angle. We refer to Barbano et al. (2022a, Appendix A) for a derivation.

Departing from the exact Gaussian setting, we turn to approximating the acquisition functions using samples. More specifically, assume we are given image samples $\boldsymbol{x}_t^1, \ldots, \boldsymbol{x}_t^k$ from the (approximate) posterior distribution, with mean prediction $\frac{1}{l}\sum_{i=1}^l \boldsymbol{x}_t^i$. To obtain a sampling based approximation of the acquisition functions Eqs. (3) and (14), note that in the Gaussian model, the term $A_\psi \Sigma_t A_\psi^\top \in \mathbb{R}^{l \times l}$ is the variance of the posterior mean observation. This term can be directly approximated from samples, i.e.

$$A_\psi \Sigma_t A_\psi^\top = \mathrm{Cov}[A_\psi \boldsymbol{x}_t^1] \approx \sum_{i=1}^k \left(A_\psi \boldsymbol{x}_t^i - A_\psi \bar{\boldsymbol{x}}_t\right)\left(A_\psi \boldsymbol{x}_t^i - A_\psi \bar{\boldsymbol{x}}_t\right)^\top. \tag{15}$$

**Uncertainty Sampling**  Uncertainty sampling aims at querying data points with the largest posterior total variance, i.e.

$$\psi_{t+1} = \arg\max_{\psi \in \Phi} \mathrm{tr}(\mathrm{Cov}[A_\psi \boldsymbol{x}_t^1]). \tag{16}$$

The variance can be analogously approximated from posterior samples using Eq. (15).

**Query by Committee**  Lastly, committee based acquisition (Seung et al., 1992) aims at taking measurements that maximize the disagreement in the measurements for candidates $\boldsymbol{x}_t^1, \ldots \boldsymbol{x}_t^k$ towards reference prediction $\hat{\boldsymbol{x}}_t$. For example, taking the average squared Euclidean norm, we choose the angle maximizing the disagreement,

$$\psi_{t+1} = \arg\max_{\psi \in \Phi} \sum_{i=1}^k \|A_\psi \boldsymbol{x}_t^i - A_\psi \hat{\boldsymbol{x}}_t\|^2 \tag{17}$$

For a Gaussian likelihood, this corresponds to the average KL between the observation distribution induced by $\boldsymbol{x}_t^i$ and $\hat{\boldsymbol{x}}_t$ respectively. Other variants such as worst-case disagreement or other divergence measures are also possible (Hino & Eguchi, 2023). We note that when taking the mean prediction as reference, $\hat{\boldsymbol{x}}_t = \bar{\boldsymbol{x}}_t$, the committee based approach reduces to maximizing the sample variance.

### D.2  EVALUATION OF ACQUISITION FUNCTIONS

In our evaluation, we compare the three acquisition function (variance, log-determinant and committee based) to a static uniform design. While we observed significant gains using all of the active learning strategies, there was no significant difference among the different acquisition functions (although it is possible to construct examples where the acquisition functions differ). This leads us to recommend the simplest, variance acquisition strategy given our current evaluation. A detailed overview of the results is given in Fig. 19

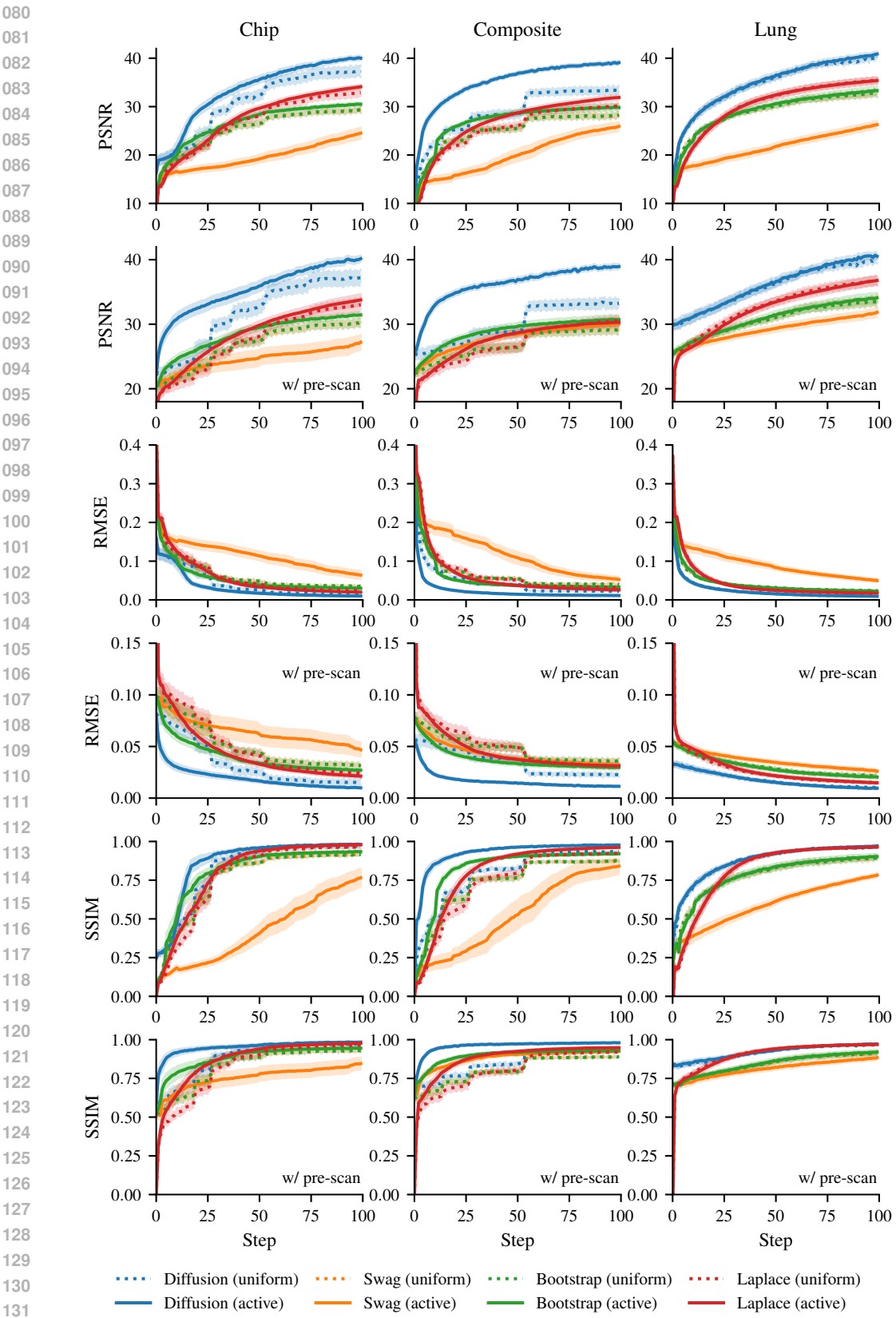

Figure 12: Results for different evaluation metrics (PSNR, RMSE, SSIM). The plots for PSNR are the same as in the main text.

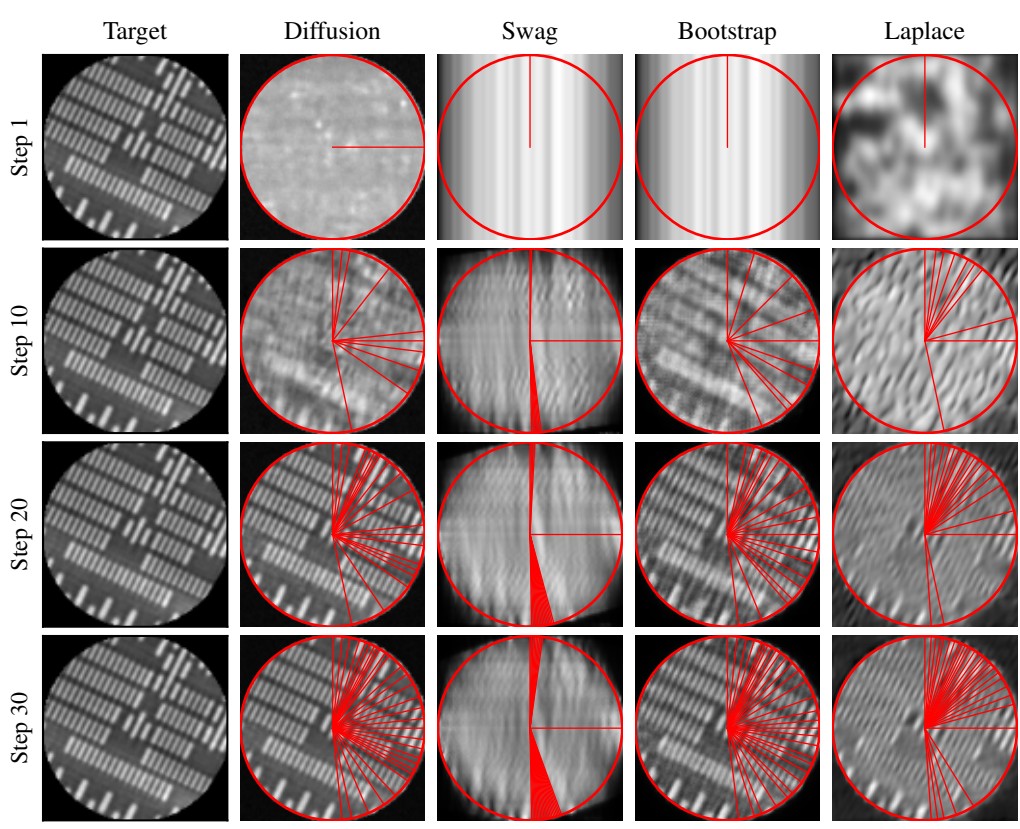

Figure 13: Qualitative results for the 'Chip' dataset ($128 \times 128$). DAL eventually focuses its attention in the direction of the chip structures and its orthogonal direction; other algorithms struggle to pick both of them. .

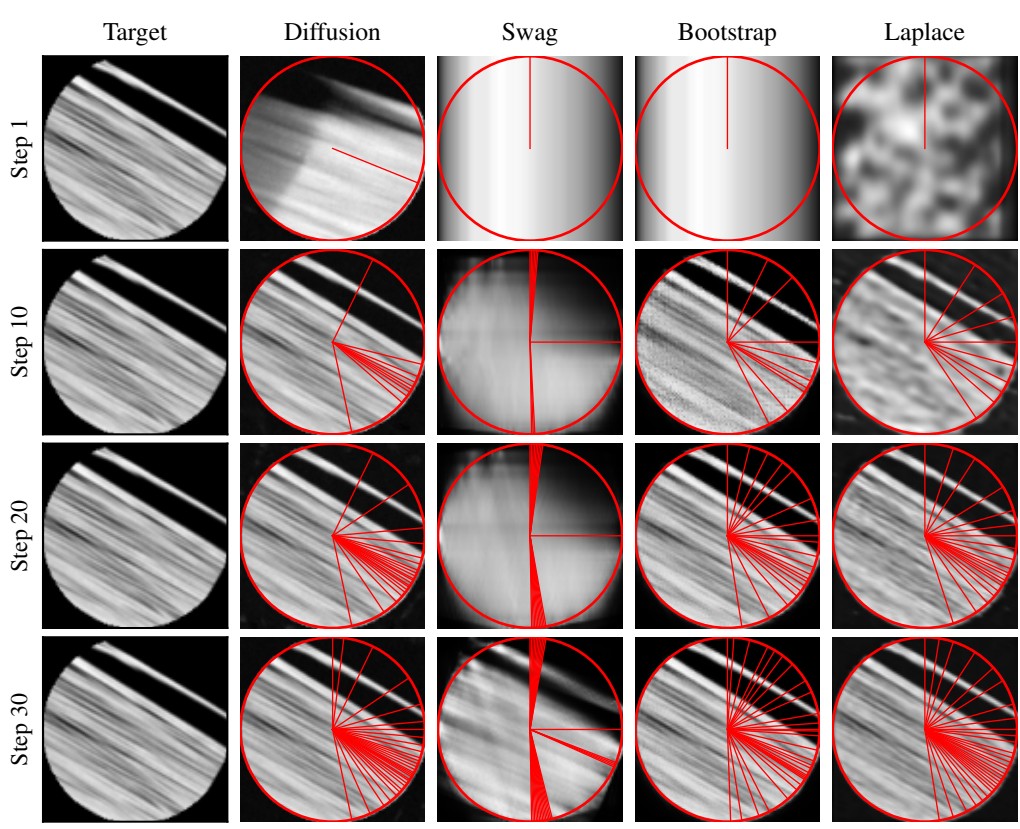

Figure 14: Qualitative results for the 'Composite' dataset ($128 \times 128$). There exists one prominent direction containing most of the information, which is quickly picked up by DAL. Other algorithms eventually catch up, but take longer to converge to this direction.

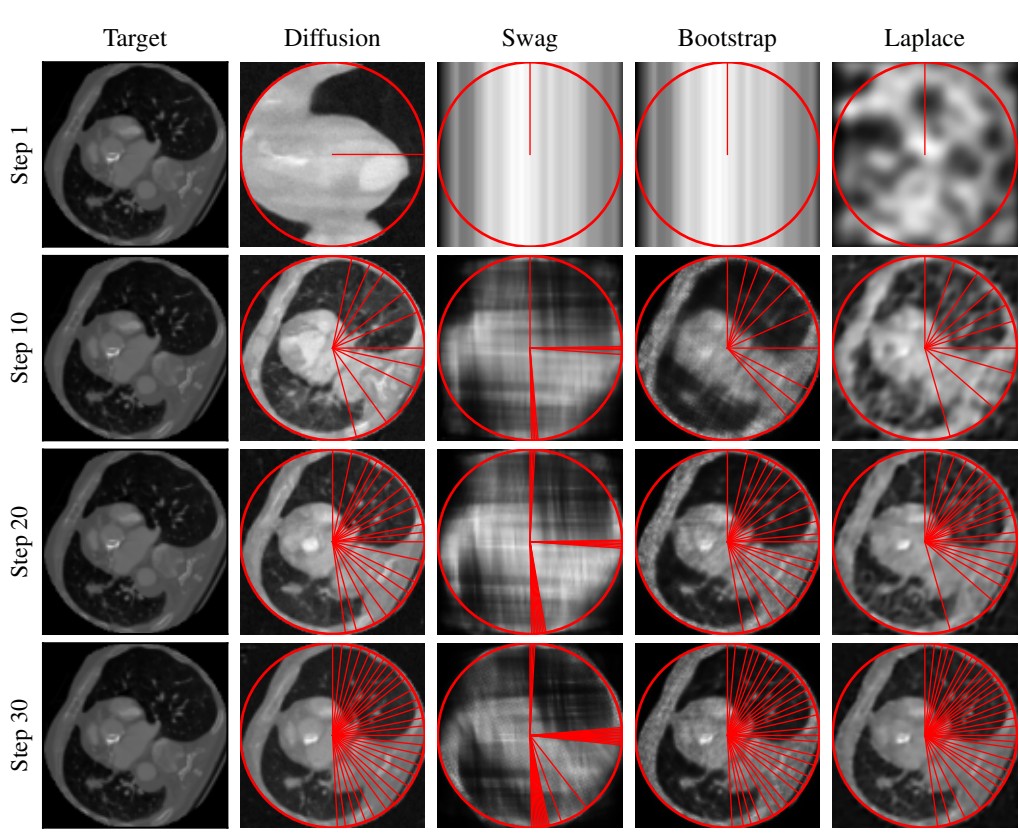

Figure 15: Qualitative results for the 'Lung' dataset ($128 \times 128$). While some structure exists in the sampling strategy with large sparsity, it quickly converges to a uniform distribution. This shows why uniform sampling is on pair with Active Learning strategies for the Lung dataset.

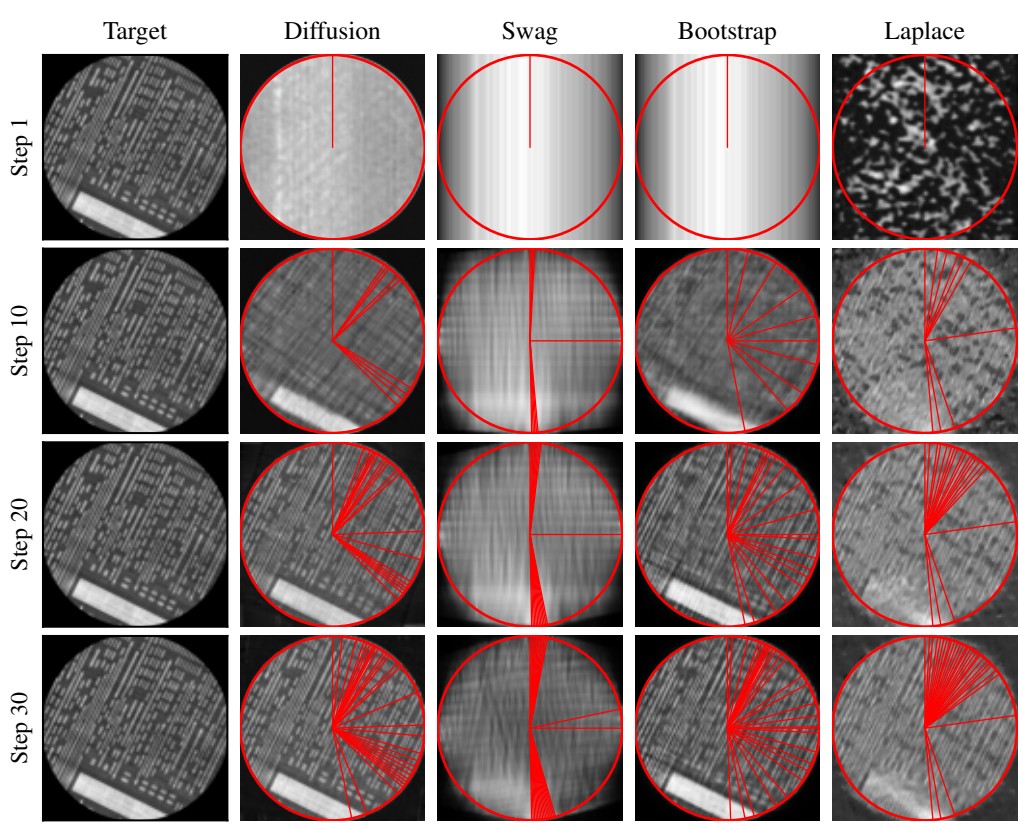

Figure 16: Qualitative results for the 'Chip' dataset ($512 \times 512$). DAL quickly focuses on the direction of the chip structures and its orthogonal direction; other algorithms struggle to pick both of them.

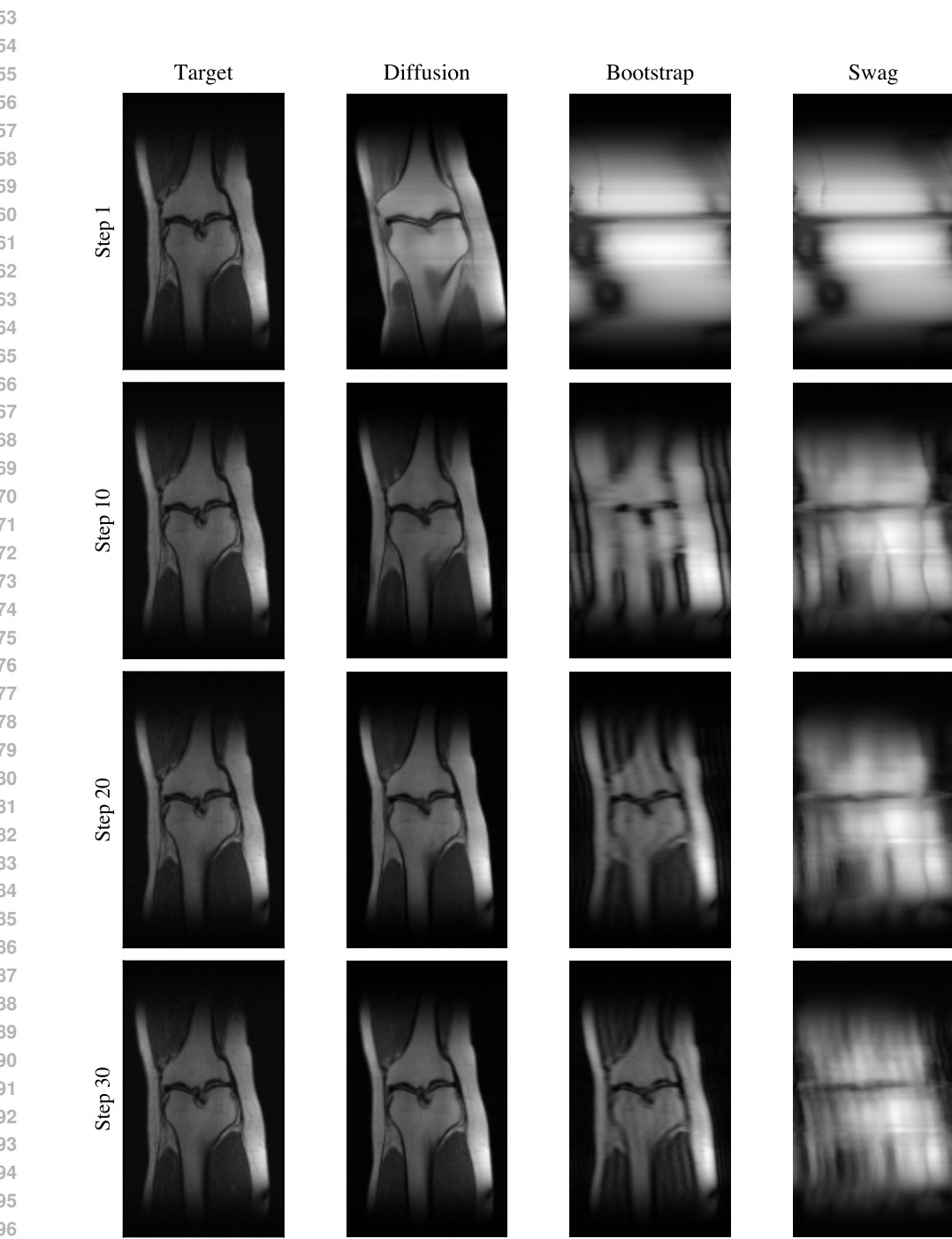

Figure 17: Qualitative results for the fastMRI dataset using active learning acquisition uncertainty sampling and 2 columns preselected.

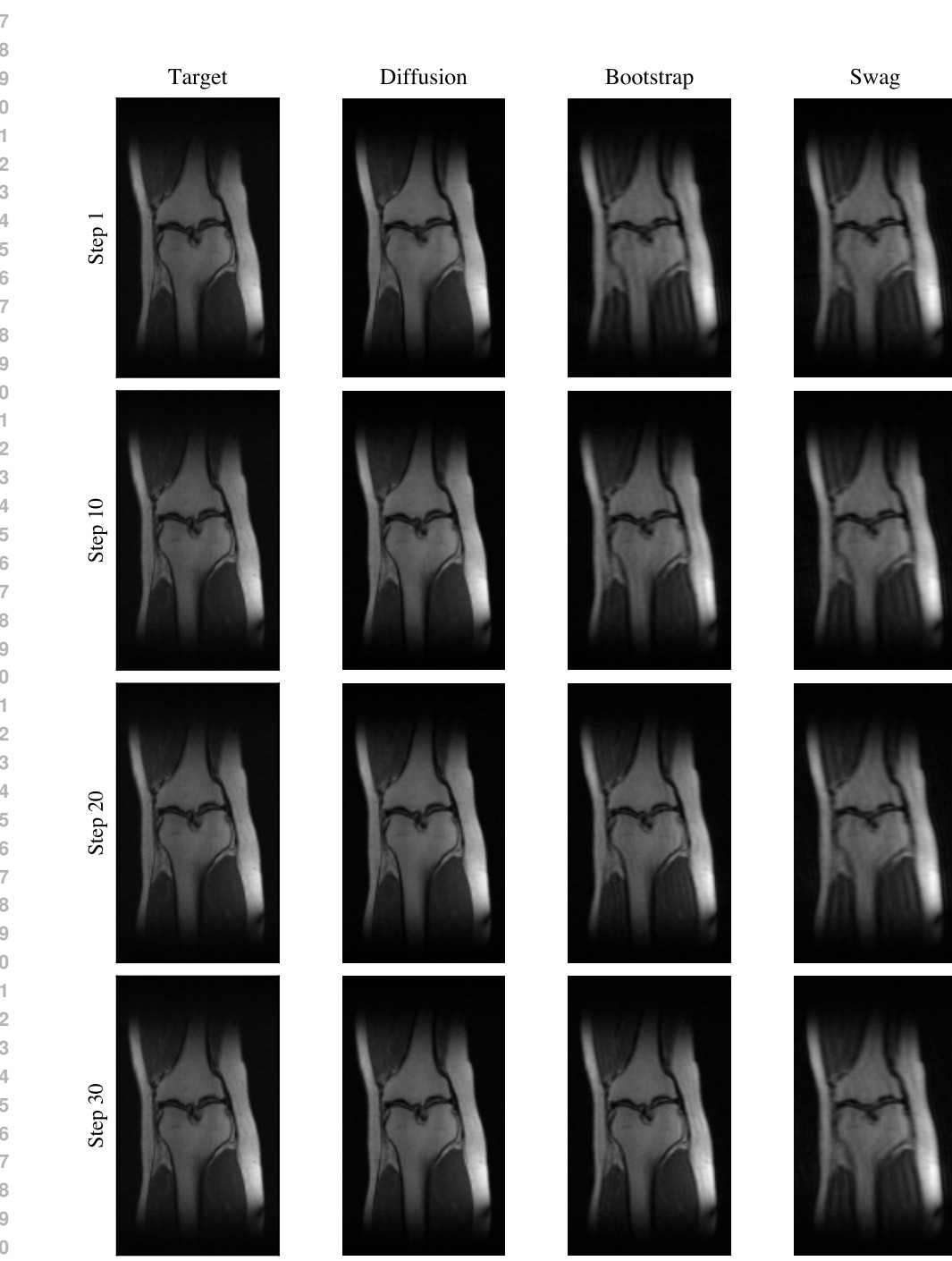

Figure 18: Qualitative results for the fastMRI dataset using active learning acquisition uncertainty sampling and 30 columns preselected.

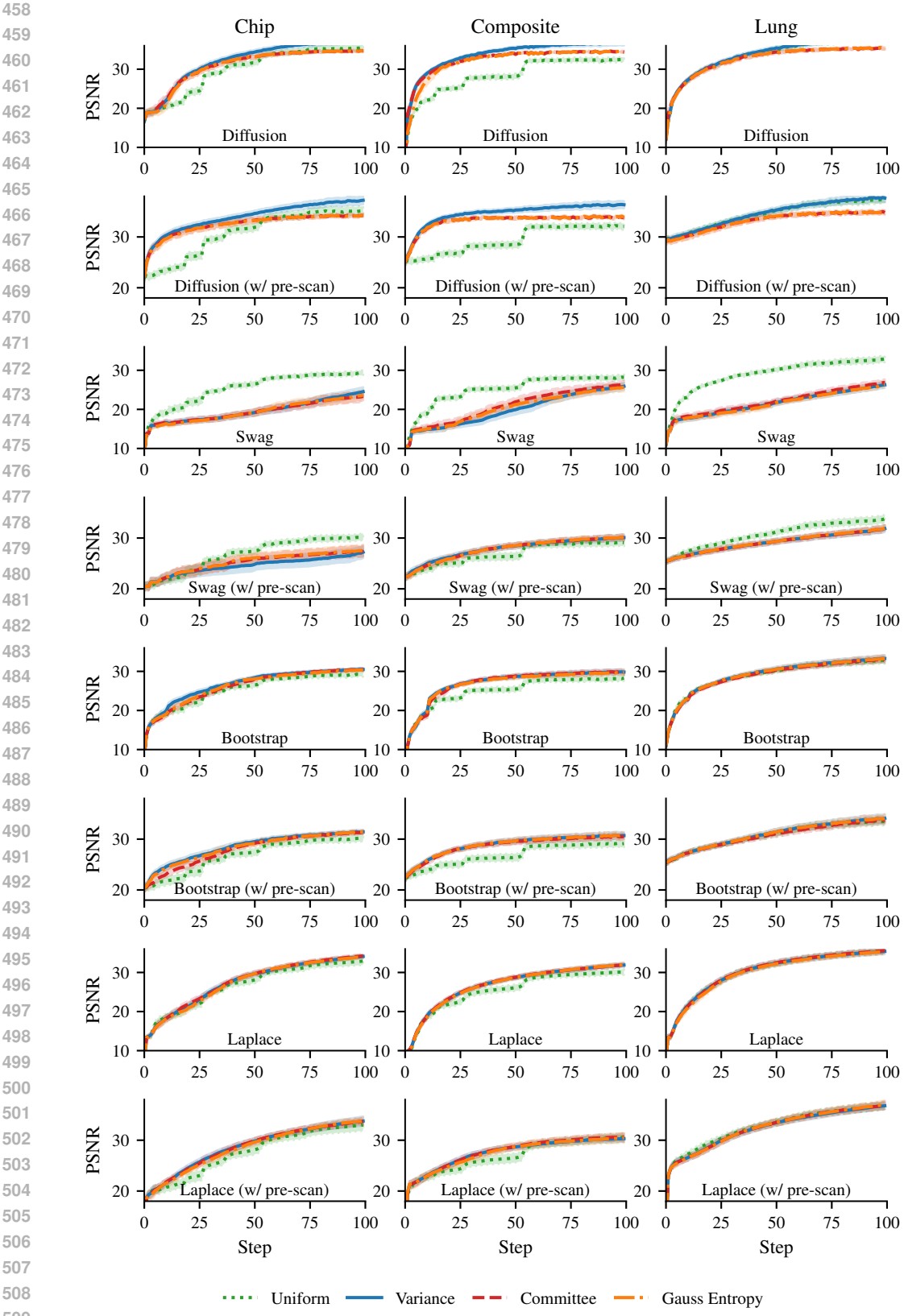

Figure 19: Results comparing different acquisition functions for each model. We show PSNR for three active acquisition strategies (Variance 3, Committee 17 and Gauss Entropy 14) and the uniform baseline. In almost all cases (except for SWAG), there is no visible difference among the different acquisition strategies.

