# OpenReview forum: "Diffusion Active Learning: Towards Data-Driven Experimental Design in Computed Tomography"
_ICLR.cc/2025/Conference — Submitted to ICLR 2025_

### Official Review · Reviewer_CoA4 · 2024-10-24

**Soundness:** 2
**Presentation:** 2
**Contribution:** 2
**Rating:** 5
**Confidence:** 3

**Summary:**

The paper proposes a framework for adaptive-sampling in the context of limited-angle X-ray/computed tomography (CT). Using measurements collected from a subset of angles, a diffusion model is used to generate approximate posterior samples. Then, the forward model is applied to each posterior sample to obtain the corresponding measurement at different angles. The uncertainty is represented as the variation in the measurements at each angle. Finally, the angle with the largest variation is selected as the next angle to collect measurements for. The paper shows that the active learning approach provides higher PSNR with fewer measurement steps compared to uniform sampling.

**Strengths:**

- The paper tackles an important problem in the field of limited-angle CT. Long-scan times and high radiation doses clearly pose hurdles in all applications from medical tests to chip analysis.
- The solution is well-motivated. By identifying the angles with the most uncertainty, the proposed method promises to select the next angle with the most information.
- The experiments demonstrate notable gains in PSNR using the active learning approach versus the uniform sampling.

**Weaknesses:**

- The contributions of the paper were not explicitly clear to me. From the experiments, there were two independent variables that were changed: 1) the method used to generate samples and 2) the use of the active learning procedure. Is the main contribution the use of a diffusion model for the sampling procedure? Or is the main contribution the active learning procedure? Or is the combination of the two the main contribution? The diffusion sampling is based on an existing approach (Song et al. 2023), and it seems like the active sampling approach is based on existing uncertainty sampling. Thus, it is difficult to see where the novelty/contribution of the paper lies. It would be helpful if you could explicitly stated the contributions in a set of bullet points in the introduction.
- The structure of the experimental section is confusing, particularly section 4.2. There is not any context as to what the methods (SWAG, Bootstrap, etc). are used for. Before introducing them, it would be helpful to identify where they are utilized in the framework. It was not clear to me until the results section that they would be substituted in for the diffusion sampling. Also, "Comparison Methods" would be a better suited title for the subsection.
- In a similarly light, the paragraph from lines 468-473 lacks context. It is unclear which Table/Figure the analysis is discussing.
- I'm not fully convinced about the practical advantage of the active sampling with the diffusion approach. As stated in the conclusion, diffusion models are inherently computationally heavy and slow. Thus, while you may need fewer measurements overall, the collection of each measurement would take much longer. For example, if it takes x times as long to choose the next angle than it does to just sample the next uniform angle, then you would want to show that your method allows you to collect at least x times fewer samples.
- In Line 300-301, it would be useful to use a different variable rather than t in order to avoid confusion with the diffusion time steps.

**Questions:**

- In Eq 3, why do you take the mean of the posterior samples first and then apply the forward operator? Would it make more sense to take the mean of the measurements (i.e. apply the forward operator first and then take the mean)?

---

> ### Author Response · Authors · 2024-11-27
>
> Dear Reviewer,
>
> Please refer to the global response for a discussion of points that was shared across the reviews, including:
> * a discussion of contributions
> * ablation attributing the effect from better reconstruction and better angle selection
>
> Further details to your specific questions are below.
>
> *"The structure of the experimental section is confusing, particularly section 4.2.
> In a similarly light, the paragraph from lines 468-473 lacks context. It is unclear which Table/Figure the analysis is discussing."*
> * Thank you, will update the experimental section with additional experiments for the final version of our paper, and address your feedback.
>
> *"I'm not fully convinced about the practical advantage of the active sampling with the diffusion approach. "*
> * As mentioned above, the main application of our technique is in synchrotron X-ray facilities, where experiments can take several hours or days. In this case, the computational overhead of DAL is negligible, and the gains in quality from AL strategies have a significant impact in beam time and in costs. We agree however that these gains are less significant in medical imaging, which, however, was never intended as the primary application of our work. In fact, one of our contributions is to show that AL strategies do not work for the lung data set, where there is little to no advantage over uniform sampling.
>
>
> *“In Line 300-301, it would be useful to use a different variable rather than t in order to avoid confusion with the diffusion time steps.”*
> * Thank you for noting this clash in notation, we will change the notation to avoid confusion.
>
> *“In Eq 3, why do you take the mean of the posterior samples first and then apply the forward operator? Would it make more sense to take the mean of the measurements (i.e. apply the forward operator first and then take the mean)?”*
>
> * The order in which the empirical expectation is computed makes no difference for linear forward models (CT, MRI).  In non-linear forward models, the order makes a difference. Taking the expectation first and then applying the forward model corresponds more closely to a committee-base method, which aims to distinguish a candidate model (the mean estimate) from alternative models via the (Gaussian) KL divergence. Applying the forward model first and then taking the expectation resembles more closely the uncertainty sampling approach, using the variance of the observations to guide the acquisition.

---

> > ### Comment · Reviewer_CoA4 · 2024-12-02
> >
> > Thank you for your detailed global and personal response to my concerns.
> >
> > After carefully reading your responses and the other reviews, I've decided to increase my score to a 5 but still cannot fully recommend the paper for acceptance for the following reasons:
> >
> > 1) I can now see the novelty and contribution of the framework better, but only by reading the related paper by Elata et al. To me, the true contribution of the paper is the development of an active acquisition procedure that utilizes posterior samples. The presentation of the paper makes it seem like the contribution is simply combining an existing posterior sampling method with an existing active acquisition method, which lacks novelty. Instead, Elata et. al. emphasize that this is a new active acquisition method that is enabled by having posterior samples. The experimentation also reflects this confusion since you compare the same active acquisition procedure with different posterior sampling approaches, suggesting that your main contribution is trying an existing active acquisition approach with a better sampler. Instead, Elata et al. compares against existing active acquisition approaches (non-posterior-sampling-based), which better demonstrates that the entire framework is novel. Overall, I think the concept is novel, but the paper does not highlight the correct contributions in the writing or in the experimentation.
> >
> > 2) The ablation study in Appendix C.3 demonstrates that most of the improvement in Fig 4 is just a result of using a better posterior sampler. Even though using the Bootstrap steps results in a slightly lower performance, this can also be attributed to having a worse posterior sampler when deciding the acquisition steps. Thus, to me, the main takeaway from the experiments is that diffusion models provide better samples, which is not surprising. This again points to the need of comparing against existing active acquisition methods in order to highlight your contribution.
> >
> >
> > Ultimately, I think the proposed framework is in fact novel, but the paper, in its current form, does not reflect the novelty in the writing or in the experimentation. The experiments should be restructured to compare against existing active acquisition methods, thus demonstrating where this paper fits in the context of active acquisition.
> >
> > Small sidenote: The use of active learning in this context is confusing to me since active learning typically refers to selecting the best data for training. I would suggest active acquisition instead to make a distinction.

---

> > > ### Author Response · Authors · 2024-12-04
> > >
> > > Dear Reviewer,
> > >
> > > Thank you for your response and additional comments.
> > >
> > > We agree that we need to emphasize the novelty and the contributions better, and we will make sure to address this for the final version of our submission. Indeed, the key insight is to develop an active learning method that utilizes posterior samples from learned diffusion prior, which enables several key advantages. Although this is already outlined in our paper (ll 60- 77), we will rephrase this to make it clearer. Thank you also for suggesting “active acquisition”, we agree this is a more descriptive term.
> > >
> > > As for the experiments, our current experiments demonstrate the difference of utilizing posterior samples for active acquisition from a learned prior compared to a dataset independent generative model. We agree that additional active learning baselines will highlight our contributions better, and we have already added two additional baselines : https://drive.proton.me/urls/Y4H5V360QR#o6THDhGTNA1j
> > >
> > > For the additional active learning baselines, notice that both RL [2] and the Active learning strategy of [1] (Wang et al.) are proposed to work with fixed-orientation datasets, and from our experiments, this seems to be a crucial assumption. We trained (and tested) with fixed orientation and independently trained and tested with random orientation. The results show a substantial gap between these two settings, where RL and [1] are massively affected by the unknown orientation of the object.
> > >
> > > [1] Ce Wang, Kun Shang, Haimiao Zhang, Shang Zhao, Dong Liang, S. Kevin Zhou. Active CT Reconstruction with a Learned Sampling Policy, Proceedings of the 31st ACM International Conference on Multimedia, October, Pages 7226–7235, 2023
> > > [2] Shen, Z., Wang, Y., Wu, D., Yang, X., & Dong, B. (2020). Learning to scan: A deep reinforcement learning approach for personalized scanning in CT imaging. arXiv preprint arXiv:2006.02420.
> > >
> > >
> > > For the ablation study, you write “Even though using the Bootstrap steps results in a slightly lower performance, this can also be attributed to having a worse posterior sampler when deciding the acquisition steps”. We exactly believe that the difference is attributed to having a better posterior sampler. The key point is that a better posterior sampler (e.g. a diffusion model) leads to better acquisition steps that are aimed at reducing the posterior variance of the predictions.

---

### Official Review · Reviewer_FMC9 · 2024-11-02

**Soundness:** 4
**Presentation:** 4
**Contribution:** 3
**Rating:** 8
**Confidence:** 4

**Summary:**

This paper utilizes a generative model which is then used in the active learning process to choose the next, most informative measurement. First, the authors train an unconditional diffusion model on a specific-dataset. In the second step, samples are generated whereby the diffusion model is conditioned on the measurements. With these samples the next measurement angle is chosen which has the highest posterior variance. According to this the total dose and acquisition time can be reduced.

**Strengths:**

* The paper contains a really good explanation of the novel approach.
* The approaches, results and limitations of already existing work is well discussed.
* Reducing the dose or measurement time during the CT measurement is an essential problem.
* Novel combination of diffusion models with active learning.
* The results are well discussed and compared to different baselines.

**Weaknesses:**

* Pre-training of the diffusion model is necessary. Further steps depend on this.
* The diffusion model is highly dependent on the trained data.
* The diffusion model could introduce undesirable biases.
* How to get the posterior distribution could be discussed in more detail.

**Questions:**

* Why are medical images not suited for this approach? In the paper it is stated because they are acquired very fast and therefore sparse but the goal is to have fewer measurements while keeping the resolution high?
* How does this model perform with samples that are slightly out of the distribution the diffusion model was trained on?
* Samples can be destroyed when a high dose or a long-time measurement is taken. How would this approach reconstruct the image? Would it automatically reduce the distortions which could be undesirable?
* Why are the smaller images first cropped and then rescaled? The distribution changes when rescaling images.

---

> ### Author Response · Authors · 2024-11-27
>
> Dear Reviewer,
>
> Thank you for the feedback and evaluation of our work. Please refer below for the answers to your questions.
>
> *Why are medical images not suited for this approach?*
> * Our results show that the benefit of Active Learning is dataset specific, i.e., it depends on the intrinsic structure of the samples. In the case of the medical images in the Lung dataset we tested, the benefit that AL has over uniform sampling is almost negligible. This is not the case for structured dataset like the composite and chip dataset studied in our paper. So while the diffusion-based reconstruction technique can be applied for medical images and will have state of the art reconstruction quality, our claim is that the results of the active acquisition will be almost indistinguishable from those of obtained with uniform sampling. For MRI the benefit of AL exists, but it is small.
> Besides the little advantage of active acquisitions in the medical datasets tested, there are also safety considerations in high-stakes medical settings that might preclude direct use of diffusion-based reconstructions. On the other hand, we claim these techniques are better suited for synchrotron facilities, where imaging times are long, and samples often have distinct structures such as the Chip and Composite data.
>
> *How does this model perform with samples that are slightly out of the distribution the diffusion model was trained on?*
> * If the samples are far from the trained distribution, the reconstructions of the diffusion model are not accurate and the method could fail. However, if samples are relatively close, even though the reconstructions might not have high PSNR at the beginning, the structure of the predicted image can guide the active learning process and still provide information on the directions to sample. In this case one would likely need more measurements to avoid hallucinations from out of distribution samples.
> Another line of research is to use the approach proposed by Barbano et al. (2023) (cited in our paper) which proposes a steerable conditional diffusion model designed to adapt to out-of-distribution scenarios in imaging inverse problems. In this case, one could trade-off some quality for robustness. This is a really interesting area of research that deserves further attention, and we believe that our paper can further spark interest in this problem.
>
> *Samples can be destroyed when a high dose or a long-time measurement is taken. How would this approach reconstruct the image? Would it automatically reduce the distortions which could be undesirable?*
> * We believe our approach can cope better with damage than other methods like FBP, as the final output is forced to lie within the training distribution. Whether this is desirable or not, depends heavily on the application and the goal of the experimenter; if the sample was damaged to the point it is no longer in the distribution, then we are back in the case discussed above.
> However, we have not benchmarked the capabilities of our model to deal with radiation damage. This is however orthogonal to the Active learning strategy proposed, and is indeed an interesting line of research of diffusion-based reconstruction methods for CT.
>
> *Why are the smaller images first cropped and then rescaled? The distribution changes when rescaling images.*
> * The datasets tested in our paper come from real tomographic reconstructions. However, the tomograms have a few thousand pixels per side, this is not a size with which we can do extensive benchmarking, and are at the edge of what can be done with modern diffusion models. Therefore, we used crops of these tomograms. The images at 512x512 have no scaling and reflect the real pixel resolution of underlying distribution. For 128x128, the 128x128 crops have a small field of view, so we took 256x256 images and downscaled them to 128x128. This indeed changes the distribution, but does so fairly for all benchmarked algorithms. This 128x128 size is mainly used to run extensive tests and have statistical significance.

---

> > ### Author Response · Authors · 2024-12-04
> >
> > Dear Reviewer,
> >
> > We hope that our previous response clarifies your questions.
> > We further briefly comment on the remaining points in your review below:
> >
> > ”Pre-training of the diffusion model is necessary. Further steps depend on this.”
> > * Yes, pre-training the diffusion model is a necessary step to learn a data-dependent prior distribution. We note however that the training is independent of the forward model or a specific reconstruction method. This is unlike many of the prior works that learn a strategy for a fixed forward model or a specific reconstruction method, and changing either requires to re-train the policy. In our case, changing the way the posterior variance is computed suffices.
> > We believe that our work could further spark interest in training a foundation model on CT and MRI images that, if trained with enough labeled data, can be used zero-shot or fine-tuned quickly to produce meaningful posterior samples.
> >
> >
> > “*The diffusion model is highly dependent on the trained data.
> > The diffusion model could introduce undesirable biases.*”
> > * We believe that this trade-off is to some extent unavoidable: To enable more efficient acquisition and reconstruction, one has to exploit prior structure about the data, which inevitably introduces a bias. On the other hand, having no bias (i.e. no prior assumptions) in the under-constraint (sparse) reconstruction settings means that a perfect reconstruction is impossible. This is the essence of the “no free lunch theorem”. The right trade-off between robustness and efficiency is application dependent (e.g. medical applications vs overview scans of composite materials), and understanding the trade-off better is an exciting direction for future work.
> >
> > “*How to get the posterior distribution could be discussed in more detail.*”
> > * “Thank you, we will revise the paper to include additional details on how to get the posterior distribution. Note that Appendix A already has a discussion of the poster diffusion sampling, please let us know if you prefer additional details in the main text, or beyond what is presented in Appendix A.

---

### Official Review · Reviewer_8GRq · 2024-11-03

**Soundness:** 3
**Presentation:** 3
**Contribution:** 3
**Rating:** 5
**Confidence:** 5

**Summary:**

The paper proposed Diffusion Active Learning that integrates a generative diffusion model with active learning to select projection angles. Based on the pretrained  unconditional diffusion model,  the proposed model using the sampled images to select the most informative next measurement.

**Strengths:**

The proposed method use half or less measurement to achieve the same performance with the compared methods.

**Weaknesses:**

(1)	Comparison with activate learning method is preferred, such as the method proposed in [1]. Please compare the reconstruction result and inference time  with use the same number of projection angle.
(2)	The experiment is performed with parallel radon transform. More complex setting, such as fan-beam or 3D Cone beam, can verify the effectiveness of the Proposed method.
(3)	The inference time is a huge disadvantage for you need n round k times sampling and n times full view projection. During, the n\times k sampling, the inversion problem cannot be avoided.  Please discuss potential ways to mitigate the computational cost. Please give  a more detailed analysis of the trade-off between computational cost and reconstruction quality.
(4)	 The improvement of the result may come from the diffusion model.  Comparison with DPS or proposed method without active loop using sparse view projection data, i.e. uniform projection angles (27,15,18 angles), is necessary.  This can  help give the explanation of the benefits of  proposed method  from the active learning component or the diffusion model

[1]Ce Wang, Kun Shang, Haimiao Zhang, Shang Zhao, Dong Liang, S. Kevin Zhou. Active CT Reconstruction with a Learned Sampling Policy, Proceedings of the 31st ACM International Conference on Multimedia, October, Pages 7226–7235, 2023

**Questions:**

(1)	Plot the distribution of selected angle of different datasets.
(2)	The shape of the objection has influence of the selected angel,
(3)	The setting of projection geometry must be given.
(4)	The definition of the notation must be given such as x^* in algorithm 1.
(5)	Testing on real projection data can verify the value of the proposed method.

---

> ### Author Response · Authors · 2024-11-27
>
> Dear Reviewer,
>
> Thank you for the feedback on our work. Please see our response to all reviewers which addresses points shared by several reviewers, in particular:
> * comparison and discussion of further baselines
> * evaluation on real MRI data and discussion
>
> Additional details to specific questions are below.
>
> *“(1) Comparison with activate learning method is preferred, such as the method proposed in [1]. Please compare the reconstruction result and inference time with the same number of projection angle."*
>
> * The mentioned paper learns a global sampling distribution that should work for all the samples, i.e., the sampling strategy is not sample-dependent. This is good for distributions with lower variability and that have always the same orientation. In many tomographic settings however (like synchrotron nano-tomography), the orientation is completely arbitrary, so learning a fixed distribution that is not sample-dependent will suffer even when presented with the same sample with an arbitrary rotation. For this reason, we chose to not add further non-adaptive baselines at this point.
>
>
> *“(2) The experiment is performed with parallel radon transform. More complex setting, such as fan-beam or 3D Cone beam, can verify the effectiveness of the Proposed method.”*
>
> * There are indeed several projection settings, and the proposed DAL strategy in principle applies to any forward model. In this paper, we decided to focus on parallel beam geometry given the importance in synchrotron X-ray experiments with parallel beam geometry, where time savings coming from AL are most dire. We have extended our experiments to work with MRI, and we could include fan-beam geometry in the final version of the paper.
>
> *“(3) The inference time is a huge disadvantage for you need n round k times sampling and n times full view projection. During, the n\times k sampling, the inversion problem cannot be avoided. Please discuss potential ways to mitigate the computational cost. Please give a more detailed analysis of the trade-off between computational cost and reconstruction quality.”*
>
> * While the inference of a single image can take from 10 to 20 seconds, the inference of k samples can be done in almost the same time by batching the inference in the diffusion model. We achieved between 20 to 30 seconds for an entire loop of the DAL algorithm. For scientific imaging in synchrotron X-ray facilities, where repositioning the sample can take several minutes, our running times can be included in real-time experimental setups.
> For faster setups, one could mitigate the computational cost by taking fewer diffusion steps in DDIM sampling, or by taking fewer consistency steps. This provides a trade-off between time and quality. Our technique can be further accelerated by choosing more than one angle at each iteration of the AL loop.In the case of MRI, we propose a new accelerated version that can help further mitigate these issues.
>
> *“(4) The improvement of the result may come from the diffusion model. Comparison with DPS or proposed method without active loop using sparse view projection data, i.e. uniform projection angles (27,15,18 angles), is necessary. This can help give the explanation of the benefits of proposed method from the active learning component or the diffusion model”*
>
> * In Figure 6 of the Appendix, we include a comparison with DPS and Hard data consistency, which shows that our method achieves the same or better PSNR for all tested sparsity settings.
> Since Figure 4 and 5 of our paper show already the significant advantage of DAL over taking uniform projection angles. These two remarks combined provide DAL with a clear advantage over DPS (or any other method) using uniform projection angles.
>
>
> *"(1) Plot the distribution of selected angle of different datasets. (2) The shape of the object has influence of the selected angel?"*
>
> * We added the visualization of the angles in Figures 13-15 in the Appendix. Note that diffusion active learning selects a non-uniform angle distribution for the chip and composite data, while choosing a close to uniform distribution on the lung data.
>
>
> *"(3) The setting of projection geometry must be given."*
> * We are currently using parallel beam geometry, and this is now explicitly mentioned in line 370 of the updated PDF.
> *"(4) The definition of the notation must be given such as x^* in algorithm 1. "*
> * Thank you, this is fixed.
> *"(5) Testing on real projection data can verify the value of the proposed method."*
> * See the additional experiments on the fastMRI dataset and the discussion above.
>
>
> [1]Ce Wang, Kun Shang, Haimiao Zhang, Shang Zhao, Dong Liang, S. Kevin Zhou. Active CT Reconstruction with a Learned Sampling Policy, Proceedings of the 31st ACM International Conference on Multimedia, October, Pages 7226–7235, 2023

---

> > ### Comment · Reviewer_8GRq · 2024-11-28
> >
> > I still believe evaluation with fan beam or cone beam geometry is necessary. Comparison with  related  activate learning method will add the value of the paper.  I believe that this work is not suitable for acceptance in its current form and recommend that the authors. I will not raise my rating.

---

> ### Author Response · Authors · 2024-11-29
>
> Dear Reviewer,
>
> We ran additional experiments on the chip, composite, and lung datasets using 2D fan-beam geometry. Please find an anonymous link to the results here:  https://drive.proton.me/urls/DV7FBVGQ3G#O68PZXnzeajY
>
> We found that diffusion active learning consistently and significantly outperformed both adaptive and non-adaptive baselines on the chip and composite datasets, while showing little advantage on the lung dataset (as also for parallel beam geometry). In other words, the results for parallel-beam geometry carry over to the fan-beam geometry, with a slightly less pronounced advantage for AL over uniform. Likely due to the fact that the information of a specific direction is spread over a larger set of angles in fan-beam geometry.
>
> We'd like to emphasise that one advantage of our method is that we can evaluate different beam geometries and experimental settings without retraining the diffusion model. This is not the case for several prior works that require training for a fixed forward model and a fixed reconstruction method.
> As discussed in the shared response, we also added evaluation on yet another forward model, MRI, evaluated on real measurements of the fastMRI dataset. We strongly believe that **evaluation on three different forward models and four different CT/MRI datasets** (one including real measurements) is a sufficient validation of our approach. Furthermore, as comparison with another active learning method, we also added a Reinforcement learning benchmark for CT in the pdf as mentioned in the shared response.
>
> We are now looking into adding the active learning baseline that you suggested, and we will post an update here shortly.

---

> > ### Author Response · Authors · 2024-12-04
> >
> > Dear Reviewer,
> >
> > Thank you again for your recommendations to improve our paper. We made an effort to implement these suggestions, and added both evaluation on fan-beam geometry and the active CT reconstruction baseline [1].
> >
> > * For the results on fan-beam geometry, see here: https://drive.proton.me/urls/DV7FBVGQ3G#O68PZXnzeajY
> > * For the additional active learning baseline see here:
> > https://drive.proton.me/urls/Y4H5V360QR#o6THDhGTNA1j
> > * Visualization of the selected angles is shown in Figures 13-15 in the Appendix
> > * In Figure 6 of the Appendix, we include a comparison with DPS and Hard data consistency
> >
> > For the additional active learning baselines, notice that both RL and the Active learning strategy of [1] (Wang et al.) are proposed to work with fixed-orientation datasets, and from our experiments, this seems to be a crucial assumption. We trained (and tested) with fixed orientation and independently trained and tested with random orientation. The results show a substantial gap between these two settings, where RL and [1] are massively affected by the unknown orientation of the object. Furthermore, the approach of [1] relies on a U-Net model used to infer the tomogram from the FBP reconstruction of the sparse sinogram. This is a known approach that has been benchmarked before (see Song et al. [2] Table 3), and where we know that diffusion-based strategies have a clear advantage. So even if their AL strategy is good, they are clearly outperformed by DAL.
> >
> > We will extend the evaluation to all datasets for the final version of our paper. Please also note that we added evaluation on the fastMRI dataset as suggested by reviewer hvxT.
> >
> >
> > Thank you again for your feedback. We believe the additional experiments and updates strengthen our contribution. We trust that these improvements address your concerns and provide a more complete perspective on the significance of our work. We would greatly appreciate it if you could revisit your assessment in light of these efforts.
> >
> >
> > [1] Ce Wang, Kun Shang, Haimiao Zhang, Shang Zhao, Dong Liang, S. Kevin Zhou. Active CT Reconstruction with a Learned Sampling Policy, Proceedings of the 31st ACM International Conference on Multimedia, October, Pages 7226–7235, 2023
> > [2] Song, Bowen, et al. "Solving inverse problems with latent diffusion models via hard data consistency." arXiv preprint arXiv:2307.08123 (2023).

---

### Official Review · Reviewer_hvxT · 2024-11-03

**Soundness:** 3
**Presentation:** 4
**Contribution:** 3
**Rating:** 6
**Confidence:** 4

**Summary:**

Context for those unfamiliar: Computed tomography (CT) acquires multiple X-ray _projection_ images of an object to reconstruct the 3D object. Due to ionizing radiation, there are significant risks associated with acquiring multiple X-ray viewing angles, leading to an undersampled ill-posed inverse problem. Many lines of work aim to reconstruct 3D CT using as few X-ray projections as possible.

Submission 10594 presents an active learning strategy to adaptively sample viewing angles most informative to the reconstruction, to reduce overall X-ray dosage. It first pretrains a diffusion model on fully sampled CTs from the same domain. Then, during inference, it uses the uncertainty of the posterior samples of the diffusion model to adaptively sample new angles.

Experiments are presented on three simulated datasets, where the proposed diffusion-based method compares favorably to other generative models.

**Strengths:**

- The submission tackles an important yet rarely-trodden inverse imaging problem.
- The submission is very open with its limitations which is an absolute breath of fresh air in modern papers. For example, L078 gives a much needed disclaimer about the risk of hallucinations from generative models in ill-posed medical image reconstruction problems. The submission’s discussion does a great job of listing limitations as well.
- Overall, the submission is very clearly and straightforwardly presented and was a very easy read.

**Weaknesses:**

I am open to changing my score and look forward to the rebuttal. As of now I see the following areas that should be addressed,

## 1. The same method was presented in Elata, et al ECCV 2024

The submission has the same idea, methods, and subject matter as [Elata, et al ECCV 2024](https://arxiv.org/abs/2407.08256). **This overlap does not affect my rating** as ICLR’s reviewer guide states that papers that came online after Jul 1 count as contemporaneous and Elata et al first appeared on Jul 11.

However, could the authors please enumerate the technical differences between the works such that readers can have clear takeaways from this paper?

For example, the acquisition function is different between the two papers, but their covariance-based acquisition function does seem to be inadvertently benchmarked in the Appendix of this submission as well and they perform identically.

## 2. Limited experiments

My biggest reservation is w.r.t. the submission’s limited experimental depth from the following aspects.

### 2.1. Missing Active CT baselines

While somewhat niche, active learning for CT reconstruction has been studied by previous works as well. For example,
- https://arxiv.org/abs/2006.02420
- https://arxiv.org/abs/2211.01670
- https://dl.acm.org/doi/10.1145/3503161.3548204

Could the authors please describe why these works were not discussed and/or benchmarked against in this submission? If it is feasible, it would be good to see experiments comparing the submission against them. Of course, it is understandable if this is not feasible given the limited discussion period.

### 2.2. Only CT experiments

As the submission itself states, nothing in the submission is particularly specific to CT and it could just as well be used for other sensor-domain reconstruction problems such as MRI. As MRI is widely used, has a clear case for acceleration (patient comfort, time costs, etc.), and MRI active learning is more widely studied than CT active learning, is there a specific reason why it is not studied in this submission?

Further, there are several reinforcement learning methods cited in the paper for MRI active learning. Could any of them be also adapted for CT active learning to form benchmarks for this submission?

### 2.3. No low-dose / sparse-view baseline(s)

The submission motivates itself by potentially reducing CT dosage. Low-dose and/or sparse-view CT reconstruction are immensely popular topics with both learned and hand-crafted priors used. However, the paper does not benchmark against any of the work within this field and instead only benchmarks against other sampling-based methods specifically constructed for this submission.

While I understand that sampling view prediction and low-dose reconstruction are somewhat orthogonal and can be combined, the method in this paper _requires_ the use of a diffusion model. This then precludes the use of useful low-dose reconstruction methods based on priors such as total variation.

Could the authors please discuss the differences between the proposed method and existing methods for low-dose reconstruction and whether regularizers such as TV can also be used in the proposed setup?

### 2.4. Only simulated data

While this is endemic across the field, the submission uses _only_ simulated synthetic X-ray projection data in its experiments, simulating it using the same exact forward model as it does in its model. As per the “inverse crime” phenomenon, this can create highly optimistic results and exaggerate differences between methods.

Within CT, there is a small set of datasets that provide both CT and raw _measured_ projection data. For example, please see:
- https://www.cancerimagingarchive.net/collection/ldct-and-projection-data/ (they provide scripts to rebin to fanbeam if necessary)
- https://www.nature.com/articles/s41597-019-0235-y
- https://www.nature.com/articles/s41597-023-02484-6

As detailed above, the paper could have also used active learning baselines for MRI and there are large datasets of real k-space measurements for MRI.

Could the authors please detail why the experiments only use simulated projections?

## 3. Technical contribution

Reductively speaking, the paper can be viewed as a combination of Hard Data Consistency (Song et al 2023) and uncertainty sampling. The submission instead proposes to use “soft” data consistency which is hard DC + early stopping, but it does not perform an ablation of this choice (please correct me if I missed it). As this is the primary technical delta, please perform an ablation if possible.

## 4. Minor
- Runtime requirements are not reported at all. As the paper is motivated by accelerating scans, it should quantify what the additional computational overhead boils down to.
- L462: “on pair” → “on par”

**Questions:**

- Could the authors please enumerate the technical differences between the submission and Elata24 such that readers can have clear takeaways from this paper?
- Could the authors please describe why active CT baselines were not discussed and/or benchmarked against in this submission?
- Why are the experiments limited to just CT if the method is generically applicable?
- Could the authors please discuss the differences between the proposed method and existing methods for low-dose reconstruction and whether regularizers such as TV can also be used in the proposed setup?
- Could the authors please detail why the experiments only use simulated projections?
- An ablation from hard to soft data consistency would be nice.

---

> ### Author Response · Authors · 2024-11-27
>
> Dear Reviewer,
>
> Thank you for your detailed evaluation. Please see our response to all reviewers which addresses the following  points in the review:
>
> * Concurrent work by Elata, et al ECCV 2024
> * Reinforcement learning baselines
> * fastMRI benchmark
> * simulated data vs real measurements
>
>
>
> Additional details to specific questions are below.
>
> *"While I understand that sampling view prediction and low-dose reconstruction are somewhat orthogonal and can be combined, the method in this paper requires the use of a diffusion model. This then precludes the use of useful low-dose reconstruction methods based on priors such as total variation. Could the authors please discuss the differences between the proposed method and existing methods for low-dose reconstruction and whether regularizers such as TV can also be used in the proposed setup?"*
>
> * We would like to point out that in our method, when doing inference using the diffusion model, we compute gradients of the consistency loss (eq (2) of our paper) to guide the diffusion model for posterior sampling. Equation (2) however can be extended by using any regularization that is beneficial for the reconstruction. We have tried adding TV and it provides a marginal gain in our tested examples. Nevertheless, we decided not to include it in our write up, as its use is orthogonal to the main message of our paper. Other regularizations can also be incorporated.
> * For our method, we opted for the use of a gradient descent approach that iteratively refines the prediction \hat{x_0} by minimizing eq (2). However, one could envision using other iterative approaches like SART, SIRT or other regularized gradient descent methods to guide the diffusion process.
>
> *"No low-dose / sparse-view baseline(s). The submission motivates itself by potentially reducing CT dosage. Low-dose and/or sparse-view CT reconstruction are immensely popular topics with both learned and hand-crafted priors used. However, the paper does not benchmark against any of the work within this field and instead only benchmarks against other sampling-based methods specifically constructed for this submission."*
>
> * Indeed, there is a plethora of sparse-view reconstruction methods used in the literature. However, Diffusion based approaches like DDRM (Kawar et al.), Diffusion Posterior Sampling (Chung et al. 2022) and Hard Data Consistency (Song et al 2023) have been already benchmarked heavily against other classical sparse-reconstruction methods, and they showed remarkable improvements in terms of reconstruction quality. Our technique builds on top of them and improves them further as can be seen in Figure 6 comparing the performance of our reconstruction method. Thus we retain by transitivity the advantage over classical sparse-view reconstruction methods. We will make sure to highlight this in the final version of the paper.
>
>
> *“Technical contribution” / “primary technical delta”*
> * Note that we added an ablation comparing soft- and hard data consistency demonstration.
> * As highlighted above, our work makes several contributions, in particular demonstrating the benefits of using a learned diffusion prior for active learning; and how this effect is dataset dependent. See the global response for a detailed discussion.
>
> *“Runtime requirements are not reported at all.”*
> * Figure 5 (right) in the original submission shows runtimes for all methods; note that this is only a qualitative assessment as performance improvements are likely possible for all methods. In the context of long data acquisition times of X-ray nano-tomography (up to several days), the cost of performing diffusion posterior sampling is offset by the improved sample efficiency, even for relatively expensive diffusion models. In addition, our proposed soft-data consistency posterior sampling is significantly faster compared to prior works (see additional experimental evaluation).

---

> > ### Comment · Reviewer_hvxT · 2024-11-30
> >
> > I thank the authors for the detailed response. I'm raising my score to a 6 as several of my initial points have been addressed.
> >
> > It is not higher as the paper exclusively trains and evaluates on simulated projections/k-space generated from reconstructed data. Several datasets provide raw measurements, including the three CT datasets I linked to in my original review and (to my limited knowledge) the multi-coil brain data in fastMRI.

---

> > > ### Author Response · Authors · 2024-12-04
> > >
> > > Dear Reviewer,
> > >
> > > Thank you for your response and additional comments.
> > >
> > > The current evaluation uses single-coil emulated measurements, which seems to be the standard to test MRI reconstruction methods. Indeed, most prior and concurrent works in active MRI and CT acquisition relies on simulated measurements [e.g., 1,2,3,4]. We looked into using multi-coil real measurements; however, that would have meant a more complex forward model and estimating the sensitivity of each coil. This is somewhat orthogonal to our current work, and it was not possible to provide such an experiment within the short amount of time.
> > >
> > > Still, we understand the desire for a more extensive and realistic evaluation, and we are considering adding an additional evaluation either based on the CT dataset [5] or the multi-coil data from the FastMRI dataset. We believe that this will be useful to better understand the robustness of the proposed method and the sim-to-real gap.
> > >
> > > [1] https://arxiv.org/abs/2007.10469  (single-coil data)
> > >
> > > [2] https://dl.acm.org/doi/10.1145/3503161.3548204 (simulated sinograms)
> > >
> > > [3] https://arxiv.org/abs/2006.02420  (simulated sinograms)
> > >
> > > [4] https://arxiv.org/pdf/2407.08256 (single-coil data)
> > >
> > > [5] https://zenodo.org/records/8014907

---

### Author Response · Authors · 2024-11-27

Dear Reviewers,

We would like to thank you for your time and the valuable feedback. Several reviewers had raised questions about the contributions and benchmarks, which we answer below. Please refer also to the updated pdf for additional analysis of our experimental evaluation, including a RL baseline and evaluation on the fastMRI dataset. We also comment on the concurrent work by Elata et al (2024).

## Summary of Contributions

* We introduce diffusion active learning, a novel approach that **combines** diffusion posterior sampling and active learning for angle selection in CT (and row selection in MRI, see below)
* To the best of our knowledge, this is **one of the first works to demonstrate the benefit of using diffusion models posterior sampling in combination with active learning**. In particular, we demonstrate that the advantage is **not** purely additive (i.e. cannot be attributed to better angle selection or better reconstruction alone), but the best result is obtained precisely by the combination of both (see Figure 11 and Section C.3 in the updated PDF).
* Moreover, our work highlights how “hallucinations” of the **diffusion model captures the variance in the estimation**, which is essential for the active learning process in the early stages; at the same time,  **diffusion posterior sampling ensures data-consistency** so that with enough measurements, the final reconstruction contains no unwanted hallucinations.
* We demonstrate that the **efficacy of active learning is dataset dependent**: Our evaluation shows clear gains on the composite and chip data, and no gains on the lung data set with the CT model. This on its own is an important observation and has not been emphasized in earlier works. It should not be overlooked as the potential gains for any active learning method are tied to the particular data distribution, and the forward process governing the observations. In some scenarios, such acquisition (e.g., CT on lung data) does not allow for efficient adaptive sampling, and one should not expect to have a method that universally works on general data distributions under constrained acquisition setups.
* We demonstrate a **significant computational advantage of using soft-data consistency** instead of hard data consistency as a diffusion posterior sampling approach (see additional experimental evaluation below, and Figure 6 in the updated PDF).

---

> ### Author Response · Authors · 2024-11-27
>
> ## Concurrent Work by Elata et al, ECCV 2024
>
> We thank Reviewer hvxT for raising awareness about the concurrent work by Elata et al (ECCV 2024). We carefully went over this work, and indeed, the proposed method by Elata et al. is similar to ours, and coincides in the case of linear forward models. However, there are several technical differences, also in the presentation and evaluation:
>
> * Elata et al. motivate their approach via a PCA decomposition motivated by the linear inverse problem setup, whereas our paper derives the proposed algorithm as a general instance of uncertainty sampling. In the linear case, the resulting acquisition strategies effectively coincide. However, our proposed maximum variance acquisition function provides a perspective that also applies to non-linear forward models, while it is perhaps less clear how to extend the linear PCA to non-linear forward models.
> * Their work uses DDRM-based inference methods, which have been outperformed by new conditional sampling approaches for inverse problems like Diffusion Posterior Sampling (Chung et al. 2022) and Hard Data Consistency (Song et al 2023). The sampling method we propose in this paper, matches or surpasses both DPS and Hard Data Consistency in terms of speed and quality for CT reconstructions (see Figure 6 with additional experimental evaluation).
> * Our evaluation focuses on CT reconstruction, while we found the experiments of Elata et al on CT data to be very limited. The results reported by Elata et al on CT data only show a single example, partially contradicting our evaluation on a larger set of lung scans, where we found no gains of using active learning compared to a uniform allocation. However a single example cannot be taken as representative of the whole data distribution. But even in that single example, we remark that the results by Elata et al report only a small gain compared to the uniform allocation (which can easily vanish when evaluating on a larger set of images). The reported numbers also do not reflect the variability of the random design baseline - any design has equal probability under the uniform distribution and clearly some will perform better and other worse.
> * The benchmarks of Elata et al on CT and MRI data are difficult to interpret, as they do not report confidence estimates / standard errors and less fidelity in terms of the number of angles/rows selected.
> * Elata et al. do not report findings that indicate that expected gain from active learning are strongly dataset dependent.
> * Elata et al. do not report the same baselines as we do, for example the Laplace approximation, which has been proposed in the context of CT before: https://arxiv.org/abs/2207.05714
> * Our work also compares different acquisition functions.
> * Our new accelerated MRI inference provides much faster inference than that of DDRM.

---

### Author Response · Authors · 2024-11-27

## Additional Experimental Evaluation

### Better hyperparameters for diffusion posterior sampling with soft-data consistency

While re-evaluating our experimental results and performing additional ablations, we found that our implementation of soft-data consistency was using suboptimal hyperparameters which caused the PSNR to plateau after around 30-50 steps at a PSNR of around 32-35; the pdf reports now the **updated results showing larger gains for the diffusion active learning**, in particular for a large number of measurement angles. The conclusions remain the same, and the performance for very sparse data as well. The technical difference is to ensure that the data consistency is performed as a last step in the denoising update (previously, the last step in the denoising pipeline was a diffusion denoising step, adding unnecessary noise to the final diffusion samples).


### Additional baselines on reported experiments (e.g., RL)

Several reviewers where asking about RL based baselines; we initially did not consider RL methods as baselines for several reasons: First, training RL algorithms requires a full forward simulation of the data pipeline and is specific to the reconstruction method used; in addition, training an RL policy in combination with the diffusion model leads to a computational overhead, rendering the approach effectively unfeasible. In comparison, diffusion models can be trained “offline” on existing reconstructions, and do not require access to the forward model. Second, training RL algorithms is generally considered to be very sensitive to hyper-parameter choices, while diffusion models for CT reconstructions are much better understood. Third, RL methods are “black box” as they just output an angle, whereas posterior samples from the diffusion model can be more easily interpreted.

Most importantly, however, **the goal of our work is to demonstrate that diffusion models can provide a sufficiently structured prior that can be exploited for sequential data acquisition** (the exact acquisition function, e.g. variance, entropy of committee-based, is secondary, see our comparison of different acquisition functions). RL instead, arguably, addresses the orthogonal problem of learning an acquisition function that is effective for a specific reconstruction method. One could therefore ask if using RL to select angles for diffusion posterior reconstruction leads to better reconstructions than using uncertainty sampling; in our opinion this would likely lead to marginal gains at most; with a significant computational overhead and increased complexity of the overall approach. Diffusion-posterior sampling strategies are still orders of magnitude slower than FPB and even SART, which would blow up the training time of a RL policy trained that uses a diffusion model in its reconstruction step.

That said, we use the author implementation of Shen et al (2020, https://arxiv.org/abs/2006.02420) to add a **RL-based baseline** to our evaluation (using SART reconstruction). **Preliminary results are shown in Figure 5 for the first 50 steps**; we will provide a complete evaluation for all datasets for the final version of our paper.
Note that when training RL the chip dataset with a **fixed rotation**, the reconstructions initially show a small advantage w.r.t. diffusion active learning, but saturate quickly below the diffusion approach. When training the RL approach on arbitrarily rotated images (as we did for the diffusion approach), the RL approach is significantly worse than diffusion active learning.
This shows that this RL strategy is better at learning a global sampling strategy that works for all samples, instead of a sample-dependent strategy as done with DAL.


We are aware of further baselines that are reported for MRI reconstructions, however due to the limited time we could not include them for the rebuttal (e.g. https://arxiv.org/abs/2211.01670 does not provide code). Other baselines such as https://dl.acm.org/doi/10.1145/3503161.3548204 learn a global sampling distribution jointly for all the samples, i.e., the sampling strategy is not sample-dependent. This is good for distributions with lower variability and that have always the same orientation. In many tomographic settings however (like synchrotron nano-tomography), the orientation is completely random, so learning a fixed distribution that is not sample-dependent will suffer even when presented with the same sample with an arbitrary rotation.

---

> ### Author Response · Authors · 2024-11-27
>
> ### MRI Setting and fastMRI data
>
> While we decided to focus on CT for this paper, our technique is general enough to work with linear and non-linear inverse problems. In particular, the technique directly extends to the MRI setting. **The updated paper includes preliminary evaluation on the fastMRI dataset**. Like in the CT setting, we found that diffusion active learning outperforms other generative models that we consider, and non-adaptive baselines. See Appendix B in the updated pdf. We will add a RL-based baselines for the final version of our paper.
>
> We also tested on the same datasets that we used for CT, using a synthetic MRI forward model to produce the observations. We show that DAL outperforms the other active learning strategies on our test datasets (these results are currently not included in the paper as evaluating the MRI model on CT data is not realistic). However, we also noticed the dependency on the structure of the dataset, where the gap between uniform sampling and DAL sampling narrows for non-heavily structured datasets, in this case the Lung dataset.
>
> In Appendix B of the updated pdf, we propose a new technique to accelerate MRI reconstructions using diffusion-based posterior sampling. This technique improves speed by an additional factor of 4x when compared with our baseline gradient descent strategy (see Figure 9), which is already faster than DPS and Hard Data Consistency (see Figure 6).
>
>
> ### Ablation: Is the advantage of diffusion active learning due to the reconstruction method or better angle selection?
>
> Reviewers CoA4 and 8GRq raised **the question if the reported gain is due to better angle selection or due to better reconstruction, or a combination of both**. To answer this question, we **re-evaluated the sequence of angles selected by Bootstrap approach, and performed the reconstruction using the diffusion model**. As expected, this improves the PSNR of the reconstruction, but still distinctly below the quality achieved by diffusion active learning. This showcases that the best reconstruction is achieved by the combination of both: The diffusion model captures the data distribution, and the angles selected by diffusion active learning exploit the data distribution in a way that a distribution-independent approach cannot; Bootstrap and Swag use only information obtained from the current sample, and therefore intuitively cannot “reason” about the posterior distribution as the diffusion model does. Note also that the gains are not purely from better angle selection, as there is a significant gap between uniform and active selection on the composite and chip data.
>
> ### Ablation of our Soft Consistency vs Hard Data Consistency
>
> To highlight one of our technical contributions better, we performed an **ablation of soft vs hard data consistency**. Refer to Figure 6 and the extended discussion in subsection "Soft Data Consistency and Early Stopping" inAppendix A. While both our method and Hard Data Consistency (Song et al 2023) achieve similar PSNR as a function of sparsity, our method is more efficient in terms of computation time and number of gradient steps, with up to 5x speed up when many measurements are present.
>
> ### Using Simulated Data vs Real Measurements
>
> * Fortunately, forward models of CT and MRI are very well understood and the quality of several reconstruction algorithms depends heavily on their accuracy. Thus, there is little "unfair" advantage in generating synthetic projections. In fact, for the FastMRI dataset, the difference between real measurements and synthetic projections produced by our forward model is always less than 1e-7.
> * One may also argue that the “inverse crime” problem is committed already in real data sets (like FastMRI), as the provided “ground-truth” is a reconstruction from the provided measurements assuming a forward model. Hence, it is not surprising that the measurements under the synthetic forward models match the real measurements extremely well. Therefore, this problem cannot be avoided completely even when working with real data.
> * This is a common practice in similar seminal works like Diffusion Posterior Sampling (Chung et al. 2022) and Hard Data Consistency (Song et al 2023). Synthetic data guarantees the existence of a ground truth that otherwise can introduce biases during testing. We believe that our techniques should be tested with real data, and we currently collaborate with researchers at a synchrotron facility to test this algorithm with real measurements. However, we are confident that the results on synthetic data showcase the benefits of our method.

---

### Author Response · Authors · 2024-12-04

Dear Reviewers,

We'd like to briefly summarize the additional evaluation that we are providing based on the feedback of the reviews. We will add these to the final version of our paper.

Additional baselines and ablation:
* Active CT [Wang et al; 23] and RL baseline [Shen et al (2020]: Preliminary results are here: https://drive.proton.me/urls/Y4H5V360QR#o6THDhGTNA1j  The final version of our paper will contain evaluation on all datasets.
* Evaluation using fan beam geometry: https://drive.proton.me/urls/DV7FBVGQ3G#O68PZXnzeajY
* Evaluation fastMRI single-coil knee data: Appendix B
* Ablation study for soft data consistency:  Figure 6 and subsection "Soft Data Consistency and Early Stopping" in Appendix A
* Ablation angle selection vs reconstruction: Appendix C.3
* Visualization of selected angles: Figures 13-15 in the Appendix

We will also revise the text to better highlight our contributions, and discuss the concurrent work of Elata et al (for a detailed discussion of differences, see our earlier response below).

---

### Meta-Review · Area_Chair_DdND · 2024-12-17

**Metareview:**

This paper presents a new framework for CT angle selection. It is the first to combine diffusion models with active learning for this task, which has a certain degree of innovativeness. However, as the reviewers pointed out, the paper does not clearly describe its contributions and technical details, and the description of the experiments is also unclear, with some design lacked reasonable explanations. The presentation of experimental results also appears insufficient.

Therefore, I believe the paper is not yet ready for publication at ICLR.

**Additional Comments On Reviewer Discussion:**

Tree out of four reviewers gave low scores, which were only raised to 6 or 5 after thorough discussions with the authors, indicating that the paper’s readability needs improvement.

---

### Decision · Program_Chairs · 2025-01-22

Reject